

**The environmental and evolutionary history of Lake Ohrid**
**(FYROM/Albania): Interim results from the SCOPSCO deep**
**drilling project**
**Bernd Wagner[1], Thomas Wilke[2], Alexander Francke[1], Christian Albrecht[2],**
**Henrike Baumgarten[3], Adele Bertini[4], Nathalie Combourieu-Nebout[5],**
**Aleksandra Cvetkoska[6], Michele D´Addabbo[7], Timme H. Donders[6], Kirstin**
**Föller[2], Biagio Giaccio[8], Andon Grazhdani[9], Torsten Hauffe[2], Jens Holtvoeth[10],**
**Sebastien Joannin[11], Eci Jovanovska[2], Janna Just[1], Katerina Kouli[12], Andreas**
**Koutsodendris[13], Sebastian Krastel[14], Jack H. Lacey[15,16], Niklas Leicher[1],**
**Melanie J. Leng[15,16], Zlatko Levkov[17], Katja Lindhorst[14], Alessia Masi[18], Anna M.**
**Mercuri[19], Sebastien Nomade[20], Norbert Nowaczyk[21], Konstantinos**
**Panagiotopoulos[1], Odile Peyron[11], Jane M. Reed[22], Eleonora Regattieri[1,8], Laura**
**Sadori[18], Leonardo Sagnotti[23], Björn Stelbrink[2], Roberto Sulpizio[7,24], Slavica**
**Tofilovska[17], Paola Torri[19], Hendrik Vogel[25], Thomas Wagner[26], Friederike**
**Wagner-Cremer[6], George A. Wolff[27], Thomas Wonik[3], Giovanni Zanchetta[28],**
**Xiaosen S. Zhang[29]**
[1] Institute of Geology and Mineralogy, University of Cologne, Cologne, Germany
[2] Department of Animal Ecology & Systematics, Justus Liebig University Giessen, Giessen,
Germany
[3] Leibniz Institute for Applied Geophysics (LIAG), Hannover, Germany
[4] Dipartimento di Scienze della Terra, Università di Firenze, Firenze, Italy
[5] CNRS UMR 7194, Muséum National d'Histoire Naturelle, Institut de Paléontologie
Humaine, Paris, France
[6] Palaeoecology, Department of Physical Geography, Utrecht University, Utrecht, The
Netherlands
[7] Dipartimento di Scienze della Terra e Geoambientali, University of Bari, Bari, Italy





[8] Istituto di Geologia Ambientale e Geoingegneria – CNR, Rome, Italy
[9] Faculty of Geology and Mineralogy, University of Tirana, Albania
[10] School of Chemistry, University of Bristol, Bristol, U.K.
[11] CNRS UMR 5554, Institut des Sciences de l'Evolution de Montpellier, Université de
Montpellier, Montpellier, France
[12] Faculty of Geology and Geoenvironment, National and Kapodistrian University of
Athens, Athens, Greece
[13] Paleoenvironmental Dynamics Group, Institute of Earth Sciences, Heidelberg University,
Heidelberg, Germany
[14] Institute of Geosciences, Christian-Albrechts-Universität zu Kiel, Kiel, Germany
[15] Centre for Environmental Geochemistry, School of Geography, University of
Nottingham, Nottingham, UK
[16] NERC Isotope Geosciences Facilities, British Geological Survey, Keyworth,
Nottingham, UK
[17] University Ss Cyril and Methodius, Institute of Biology, Skopje, Macedonia
[18] Dipartimento di Biologia Ambientale, Università di Roma "La Sapienza", Rome, Italy
[19] Dipartimento di Scienze della Vita, Laboratorio di Palinologia e Paleobotanica,
Università di Modena e Reggio Emilia, Modena, Italy
[20] Laboratoire des Sciences du Climat et de l'Environnement, UMR 8212,
CEA/CNRS/UVSQ et Université Paris-Saclay 91198 Gif-Sur-Yvette, France
[21] Helmholtz Centre Potsdam, GFZ German Research Centre for Geosciences, Potsdam,
Germany
[22] Geography, School of Environmental Sciences, University of Hull, Hull, UK
[23] Istituto Nazionale di Geofisica e Vulcanologia, Rome, Italy
[24] IDPA-CNR, via M. Bianco 9, Milan, Italy
[25] Institute of Geological Sciences & Oeschger Centre for Climate Change Research,
University of Bern, Bern, Switzerland



[26] The Lyell Centre, Heriot-Watt University, Edinburgh, UK
[27] Department of Earth, Ocean and Ecological Sciences, School of Environmental Sciences,
University of Liverpool, Liverpool, UK
[28] Dipartimento di Scienze della Terra, University of Pisa, Pisa, Italy
[29] Institute of Loess Plateau, Shanxi University, Taiyuan, China
Correspondence to: B. Wagner (wagnerb@uni-koeln.de)

## Abstract

This study reviews and synthesises existing information generated within the SCOPSCO
("Scientific Collaboration on Past Speciation Conditions in Lake Ohrid") deep drilling
project. The four main aims of the project are to infer (i) the age and origin of Lake Ohrid
(Former Yugoslav Republic of Macedonia/Republic of Albania), (ii) its regional
seismotectonic history, (iii) volcanic activity and climate change in the central northern
Mediterranean region, and (iv) the drivers of biodiversity and endemism. The Ohrid basin
formed by transtension during the Miocene, opened during the Pliocene and Pleistocene, and
the lake established *de novo* in the still relatively narrow valley between 1.9 and 1.3 Myr ago.
The lake history is recorded in a 584 m long sediment sequence, which was recovered within
the framework of the International Continental Scientific Drilling Program (ICDP) from the
central part (DEEP site) of the lake in spring 2013. To date, 50 tephra and crypto-tephra
horizons have been found in the upper 460 m of this sequence. Tephrochronology and tuning
biogeochemical proxy data to orbital parameters revealed that the upper 247.8 m represent the
last 637 kyr. The multi-proxy dataset covering these 637 kyr indicates long-term variability,
with a change from cooler and wetter to drier and warmer glacial and interglacial periods
around 300 ka. Short-term environmental change caused, for example, by tephra deposition or
the climatic impact of millennial-scale Dansgaard-Oeschger and Heinrich events are
superimposed on the long-term trends. Evolutionary studies on the extant fauna indicate that
Lake Ohrid was not a refugial area for regional freshwater animals. This differs from the





surrounding catchment, where the mountainous setting with relatively high water availability
provided a refugial area for temperate and montane trees during the relatively cold and dry
glacial periods. Although Lake Ohrid experienced significant environmental change over the
last 637 kyr, preliminary molecular data from extant microgastropod species do not indicate
significant changes in diversification rate during this period. The reasons for this constant rate
remain largely unknown, but a possible lack of environmentally induced extinction events in
Lake Ohrid and/or the high resilience of the ecosystems may have played a role.
**1   Introduction**
Systematic limnological studies started in the early 20th century and were first carried out in
Europe, for example, at Lake Geneva (e.g., Forel, 1901), a number of lakes in Germany (e.g.,
Thienemann, 1918), and at Lake Ohrid on the Balkan Peninsula (reviewed in Stanković,
1960). These initial studies focused on hydrological data, such as temperature, dissolved
oxygen and bottom morphology, and on biological data, such as the distribution and ecology
of lake biota. Analytical and technological advances in the following decades facilitated a
more comprehensive understanding of the interactions between catchment dynamics,
hydrology, and the living world of lakes. This led to the establishment of new institutions,
such as the Hydrobiological Institute at Lake Ohrid in 1935 (Stanković, 1960).
Besides analyses in extant lakes, early scientists were also interested in studying past changes
in lake systems, and paleolimnology, a sub-discipline of limnology, was established in the
1920s. This field started with the collection of sediment cores from lakes to interpret
stratigraphic data on plant and animal fossils as a record of the lake's history (National
Research Council, 1996). Particularly with the establishment of radiometric dating methods in
the 1950s and 1960s, paleolimnological studies developed into a powerful tool for long- and
short-term reconstructions of the climatic and environmental history of lakes and their
catchments.
One of the most important developments in paleolimnological work has been the formation of
a multi-national continental drilling program – the International Continental Scientific



Drilling Program (ICDP). The 'Potsdam Conference', conducted in 1993, defined the
scientific and management needs for the ICDP and declared Lake Ohrid, Europe's oldest
freshwater lake, as an ICDP target site.
One of the most outstanding characteristics of Lake Ohrid, besides its presumed old age, is its
high degree of endemic biodiversity. With more than 300 described eukaryotic endemic taxa
(Föller et al., 2015), Lake Ohrid belongs to the most biodiverse ancient lakes, i.e., lakes that
have continuously existed for >100 kyr (Albrecht and Wilke, 2008). If its surface area is taken
into account, it may have the highest endemic biodiversity amongst all lakes worldwide.
Though Lake Ohrid has long been considered to be of Tertiary age, estimates vary
considerably, between ca. 2 and 10 Ma (reviewed in Albrecht and Wilke, 2008). Likewise, its
limnological  origin  remains  poorly  understood,  and  hypotheses  vary  between
paleogeographical connection to former marine or brackish water systems, or *de novo*
formation from springs and/or rivers (see also Albrecht and Wilke, 2008 for further
information and references therein).
The unique characteristics of Lake Ohrid, together with the lack of knowledge regarding its
origin, precise age, and limnological/biological evolution, provided the main motivation to
establish an international scientific deep drilling project. Its continuous existence over a long
timescale together with an extraordinary degree of endemic biodiversity made Lake Ohrid an
ideal 'natural laboratory' to study the links between geological and biological evolution and
to unravel the driving forces of speciation, leading to the interdisciplinary project 'Scientific
Collaboration on Past Speciation Conditions in Lake Ohrid' (SCOPSCO). The four major
aims of the SCOPSCO project are to (i) obtain more information on the age and origin of
Lake Ohrid, (ii) unravel the regional seismotectonic history including effects of major
earthquakes and associated mass-wasting events, (iii) obtain a continuous record containing
information on Quaternary volcanic activity and climate change in the central northern
Mediterranean region, and (iv) evaluate the influence of major geological events on biotic
evolution and the generation of the observed extraordinary degree of endemic biodiversity
(Wagner et al., 2014). Based on several site surveys and studies conducted between 2004 and
2011, an ICDP drilling campaign at Lake Ohrid was carried out in spring 2013 using the
'Deep Lake Drilling System' (DLDS) from the 'Drilling, Observation and Sampling of the



Earths Continental Crust' (DOSECC) consortium. In total, more than 2100 m of sediments
were recovered from four drill sites, with a maximum penetration of 569 m below lake floor
(blf) at the main drill site (DEEP) in the central part of Lake Ohrid (Fig. 1).
Subsampling and analyses are ongoing, but initial, detailed results of geological and
biological investigations of the upper 247.8 m (637 ka) of the DEEP sediment sequence and
newer results from biological studies on the extant fauna of Lake Ohrid were recently
published in a special issue in the journal 'Biogeosciences' ("Integrated perspectives on
biological and geological dynamics in ancient Lake Ohrid", edited by Wagner et al.). The aim
of this paper is to review and synthesise the results of the 14 individual papers of this special
issue and to complement them with information from former and new studies in order to
provide a comprehensive overview on progress towards achieving the four main aims defined
for SCOPSCO.
**2 Site information**
Lake Ohrid is a transboundary lake shared between the Former Yugoslav Republic of
Macedonia (FYROM) and the Republic of Albania (Fig. 1). The lake is located at 693.5 m
above sea level (a.s.l.) and has a maximum length of 30.4 km (N-S), a maximum width of
14.7 km (W-E), a surface area of 358 km$^2$, and a tub-shaped bathymetry with a maximum
water depth of 293 m, a mean water depth of ~151 m, and a total volume of 50.7 km$^3$ (Fig. 1;
Popovska and Bonacci, 2007; Lindhorst et al., 2012). Water loss occurs by evaporation (13.0
m$^3$ s$^{-1}$) and by the artificially controlled surface outflow in the northern part of the lake, River
Crni Drim, which flows into the Adriatic Sea. Outflow rates vary between 22.0 m$^3$ s$^{-1}$
(Popovska and Bonacci, 2007) and 24.9 m$^3$ s$^{-1}$ (Matzinger et al., 2006 and references therein),
depending on seasonal and long-term variations in water level of up to ~1.5 m between 1950
and 2000 (Popovska and Bonacci, 2007). The total water loss can be averaged to ~36.5 m$^3$ s$^{-1}$
and is balanced by water input from direct precipitation, rivers, as well as surface and
sublacustrine springs. Published data of the annual precipitation in the watershed of Lake
Ohrid vary between 698.3 and 1194.0 mm yr$^{-1}$, with higher precipitation at higher altitudes
and an average of 907 mm yr$^{-1}$ (Popovska and Bonacci, 2007). The average monthly rainfall




is highest in winter, with a maximum in November and December, and lowest between June
and September. The lake level, however, is highest in June due to snowmelt input and lowest
in October and November, before the start of autumn rainfall (Popovska and Bonacci, 2007).
The seasonal and long-term variations in water budget allow only an approximation of the
water input from the various sources. Direct precipitation and river inflows (45%) as well as
surface and sublacustrine karst springs (55%) contribute to the overall water input (Matzinger
et al., 2006). The River Sateska, which was previously a direct tributary of the Crni Drim, was
artificially diverted into Lake Ohrid in 1962 and is today the largest surface river inflow with
a contribution of ~15% of the total inflow of Lake Ohrid (Matzinger et al., 2006; Poposka and
Bonnacci, 2007). The karst springs are located primarily along the eastern shoreline of the
lake (Fig. 1) and karst waters originate in almost equal proportions from mountain range
precipitation and via outflow from Lake Prespa, located ~10 km to the east and ~155 m higher
in altitude (Matzinger et al., 2006). Calculating the ratio between the volume of Lake Ohrid
(50.7 km$^3$) and its outflow (~23.5 m$^3$ s$^{-1}$) results in a theoretical water residence time of ~70
years (Matzinger et al., 2006; Popovska and Bonacci, 2007). This theoretical residence time is
reduced to ~45 years, when evaporation is taken into account and calculated with the total
water output or input (~36.5 m$^3$ s$^{-1}$). However, the real water residence time is probably much
higher, as sporadic mixing intervals or incomplete mixing, variations in wind stress, or kinetic
effects of inflow water entering may affect the lake's hydrology (Ambrosetti et al., 2003). For
example, Lago Maggiore in Italy was classified as a holo-oligomictic lake prior to 1970, when
the upper 150-200 m of the water column mixed every winter and complete mixing occurred
irregularly every few years (Ambrosetti et al., 2003). This is similar to Lake Ohrid today
(Matzinger et al., 2006) and the real residence time at Lago Maggiore is higher by a factor of
3 to 4 than the theoretical residence time (Ambrosetti et al., 2003).
Physical and chemical characteristics of Lake Ohrid have been provided in several
publications and annual reports (e.g., Watzin et al., 2002; Matzinger et al., 2006; Jordanoski
et al., 2004, 2005; Naumoski et al., 2007; Schneider et al., 2014). Average total phosphorous
(TP) concentrations of <10 mg m$^{-3}$ and Secchi depths ranging between 7 and 16 m
characterise the pelagic zone of Lake Ohrid as oligotrophic. These oligotrophic conditions
explain why bottom water oxygen concentrations of above 4 mg L$^{-1}$ are recorded even in
years without complete overturn (Matzinger et al., 2006). The surface water temperature





varies between ~25°C in summer and ~7°C in winter, while bottom water temperatures are
~6°C throughout the year. The boundary between epilimnion and hypolimnion is between 30
and 50 m, depending on the season. The pH decreases from 8.6-8.9 in surface waters to 7.9-
8.4 in bottom waters. The specific conductivity is around 200 μS cm$^{-1}$ in surface waters,
around 150 μS cm$^{-1}$ at 50-200 m water depth and increases again in deeper waters. The
concentration of Si is lowest in the trophogenic surface waters, where it is taken up by
diatoms, and increases gradually to <2 mg L$^{-1}$ in bottom waters (Stanković, 1960). The littoral
part of the lake exhibits a slightly higher trophic state (Schneider et al., 2014). These meso- to
slightly eutrophic conditions in relatively shallow waters might be due to a direct input of
nutrients from the catchment, higher temperatures, and increasing anthropogenic pollution
over the last several decades (Kostoski et al., 2010; Schneider et al., 2014). The macrophytic
flora in the littoral part of Lake Ohrid can be subdivided into different belts, with *Chara*
species in water depths between 3 and 30 m, *Potamogeton* species in shallow waters, and a
discontinuous belt of *Phragmites australis* along the shore (Albrecht and Wilke, 2008; Imeri
et al., 2010).
The vegetation in the catchment of Lake Ohrid can be categorized along altitudinal belts (cf.,
Filipovski et al., 1996; Matevski et al., 2011). Grasslands and agricultural land are
encountered in the littoral zone and the lowlands surrounding the lake, followed by forests
dominated by different species of both deciduous and semi-deciduous oaks (*Quercus cerris*,
*Q. frainetto*, *Q. petraea*, *Q. pubescens*, and *Q. trojana*) and hornbeams (*Carpinus orientalis*,
*Ostrya carpinifolia*) up to 1600 m a.s.l. Mesophilous/montane species such as *Fagus*
*sylvatica*, *Carpinus betulus*, *Corylus colurna*, *Acer obtusatum*, and *Abies borisii-regis*
dominate at higher altitudes up to 1800 m a.s.l. Due to intense grazing, the timberline is
between 1600 and 1900 m a.s.l. Reforestation is now slowly replacing the existing alpine
pasture lands and grasslands at and above this altitude (Matevski et al., 2011). Sparse
populations of several *Pinus* species, considered to be Tertiary relics, are located in the wider
region of Lake Ohrid (Sadori et al., 2016). Em et al. (1985) considered the Ohrid-Prespa
region to be a refugial area with remains of vegetation of other species (*Pinus heldreichii*,
*Quercus trojana*, *Juniperus excelsa*, *Aesculus hippocastanum*, *Genista radiata*).





The highest mountains in the Lake Ohrid watershed, which encompasses 1002 km$^2$ *sensu*
*stricto* and 2393 km$^2$ including the Lake Prespa catchment, reach 1532 m a.s.l. in the Mocra
Mountains to the west, and 2288 m a.s.l. in the Galičica Mountains to the east of the lake. The
average altitude of the Lake Ohrid watershed is 1109 m a.s.l. About 12% of its watershed is
located at an altitude above 1500 m a.s.l. (Popovska and Bonacci, 2007). Intensely karstified
Triassic limestones and Devonian siliciclastic bedrock dominate in the southeastern, eastern,
and northwestern catchment (e.g., Wagner et al., 2009; Lindhorst et al., 2015). Ultramafic
metamorphic and magmatic rocks including ophiolites of Jurassic and Cretaceous age crop
out in the west. The plains at the northern, northeastern, and southern lake shore are covered
by Quaternary sediments.

## 13   3   Material and methods

### 14   3.1   Field work

#### 15   3.1.1  Seismic and hydro-acoustic surveys

Seismic and hydro-acoustic surveys were carried out on Lake Ohrid between 2004 and 2009.
Parametric sediment echosounder profiles span >900 km in length and were collected at
operating frequencies between 6 and 12 kHz (SES-96 light in 2004 and SES 2000 compact in
2007 and 2008, Innomar Co.). These frequencies allowed up to 60 m of penetration into the
sediments at a vertical resolution of ~20 cm. Over 500 km of profiles were collected by
multichannel seismic surveys using a Mini GI Air Gun (0.2 L in 2007 and 0.1 L in 2008) and
a 16-channel 100 m long streamer. The Mini GI Air Gun operated at frequencies between 150
and 500 Hz and allowed a maximum penetration of several hundred metres at a vertical
resolution of ~2 m. A multibeam survey in 2009, using an ELAC Seabeam 1180 sonar
system, was used to acquire detailed bathymetric information of the lake floor below ~20 m
water depth. More detailed information on the technical specifications of the seismic and
hydro-acoustic systems, their settings, the location of the individual profiles, and the
operational logistics can be found in Wagner et al. (2014) and Lindhorst et al. (2015).



### 3.1.2  Coring and onsite analyses

Several gravity and piston coring campaigns were carried out from local research vessels or small floating platforms (UWITEC Co.) on Lake Ohrid between 2004 and 2011. Whereas surface sediments collected by gravity corer throughout the basin were used to reconstruct the recent settings and the most recent history of Lake Ohrid (e.g., Matzinger et al., 2007; Wagner et al., 2008a; Vogel et al., 2010c), piston cores with a maximum penetration of ~15 m blf were collected from the lateral parts of the lake, where the water depth did not exceed 150 m (e.g., Wagner et al., 2008b, 2009; Belmecheri et al., 2009; Vogel et al., 2010a, 2010b). These piston cores enabled a reconstruction of the environmental, climatic, and tephrostratigraphic history of the lake back to ~140 ka and provided fossil records of pollen (Wagner et al., 2009), molluscan faunas (Albrecht et al., 2010) and diatom floras (Reed et al., 2010).

Based on the site surveys, five primary target sites in Lake Ohrid were proposed for the SCOPSCO ICDP project. One of these sites, Lini (Co1262; Fig. 1), was cored in 2011 using a UWITEC platform and piston corer at 260 m water depth. Although the Co1262 sediment sequence reached only 10.08 m blf, this is the most complete Holocene sequence retrieved to date. Studies on the core material contributed to a better understanding of the tectonic activity (Wagner et al., 2012) and the Late Glacial to Holocene environmental history of the region (Lacey et al., 2015; Zhang et al., 2016).

The remaining four sites were cored in spring 2013 using the DLDS (Wagner et al., 2014; Francke et al., 2016). At the main site, the DEEP site in the central part of the Lake Ohrid basin, six holes (5045-1A to 5045-1F) were drilled with a maximum depth of ~569 m blf (Fig. 1) and an average distance of ~40 m between the individual holes (for details see Francke et al., 2016). In total, ~1500 m of sediment cores were recovered, cut into up to 1 m long segments, and stored in a reefer at 4°C before being shipped to the University of Cologne, Germany, for further processing.

Onsite analyses during the 2013 deep drilling campaign included borehole logging, core scanning for magnetic susceptibility, and sedimentological and palaeobiological core catcher analyses. Borehole logging was carried out with various probes at all four drill sites. The



logging tools comprised magnetic susceptibility (MS), dipmeter, resistivity, borehole
televiewer, spectral gamma ray (SGR), and sonic. While SGR was run through the drill pipe
in order to prevent caving of sediments into the drill hole, all other tools were run in 40-50 m
long open-hole sections, except for the uppermost 30 m blf, which were kept open with drill
pipes to allow re-entry of other probes. Details of the borehole logging tools, logging speed,
and vertical resolution are given in Baumgarten et al. (2015). Check shots were recorded for
hole 5045-1C, allowing a very good seismic to core correlation for the DEEP-site.
In order to determine volume-specific MS on the sediment cores and to carry out preliminary
core correlation, all cores were scanned onsite at a resolution of 2 cm with a Bartington
MS2C loop sensor (10 cm internal diameter) mounted on a multi sensor core logger (MSCL,
Geotek, UK). Smear slide analyses of core catcher material (~3 m resolution) from holes
5045-1B and 5045-1C were used for onsite diatom analyses (Wagner et al., 2014).
### 3.1.3  Biological sampling
Biological field sampling within the SCOPSCO project focused on the collection of living
invertebrates from the lake and its surroundings in order to conduct phylogenetic and
metacommunity analyses. The collection methods for gastropods followed those described in
Hauffe et al. (2011) and Schreiber et al. (2012), and included hand collecting, snorkeling,
sieving, and dredging from small boats or the research vessel of the Hydrobiological Institute
Ohrid. Samples were preserved in 80% ethanol for subsequent analyses.
In order to improve the interpretation of changes in sedimentary lipid biomarker composition,
samples from main modern terrestrial organic matter pools, i.e., soils and leaf litter, as well as
macroalgae and macrophytes (*Characeae* spp., *Cladophora* spp., *Potamogeton* spp.,
*Phragmites* spp.) were collected from the eastern and southern realm of the Ohrid Basin (for
details see Holtvoeth et al., 2016). All samples were oven-dried shortly after collection (70°C,
48 hours) and kept frozen prior to biomarker analysis.





**3.2   Laboratory work**
The geological work carried out on the gravity and piston cores from the site surveys and on
the cores obtained during the ICDP drilling campaign comprises a broad suite of different
analytical methods. It includes lithological description after core opening, measurement of the
geophysical properties, and granulometric, geochemical, mineralogical, and rock-magnetic
analyses. These analyses are carried out on whole core sections, on split core surfaces, and on
discrete samples (cf., Wilke et al., 2016) and are described in detail in several individual
publications (Matzinger et al., 2007; Wagner et al., 2008a, 2008b, 2009, 2012; Belmecheri et
al., 2009, 2010; Holtvoeth et al., 2010, 2016; Leng et al., 2010; Lindhorst et al., 2010; Matter
et al., 2010; Vogel et al., 2010a, 2010b; Lacey et al., 2015, 2016; Francke et al., 2016; Just et
al., 2016; Leicher et al., 2016). Dating of the sediment successions was mainly based on
radiocarbon dating and tephrostratigraphic and tephrochronological work. Tuning of sediment
proxies to orbital parameters, such as summer insolation and winter season length, or to other
records has only been carried out on the sediment sequence from the DEEP site (Baumgarten
et al., 2015; Francke et al., 2016; Zanchetta et al., 2016). Optical and geochemical
information was used for a correlation of the DEEP core sequences and led to a composite
profile of 584 meters composite depth (mcd) (Francke et al., 2016 and unpublished data).
Some of the sediment sequences were also studied for their fossil diatom, pollen, ostracod, or
mollusc compositions. The sample preparation for the micro- and macrofossil analyses and
the determination of the taxa are described in detail in the individual publications (Belmecheri
et al., 2009, 2010; Wagner et al., 2009, 2014; Albrecht et al., 2010; Reed et al., 2010;
Cvetkoska et al., 2016; Sadori et al., 2016; Zhang et al., 2016).
Information on interspecific relationships between Ohrid endemics and Balkan species, and
on the drivers of speciation processes and community changes was derived from extant taxa
by conducting molecular phylogenetic, lineage-through-time plot, and diversification-rate
analyses (for details see Föller et al., 2015 and references therein), as well as modeling of
community assembly processes (see Hauffe et al., 2016).



# 4 Results and discussion

## 4.1 Age and origin

### 4.1.1 Age

At the start of the SCOPSCO project, the age and origin of Lake Ohrid were poorly constrained. Previous geological and biological age estimates varied from 2 to 10 Ma (summarised in Albrecht and Wilke, 2008). Our new results allow for more precise age estimation. Based on SGR from borehole logging, MS from core logging, and total inorganic carbon (TIC) analyses on core catcher samples from the DEEP site, and by comparing these data with global climate records, such as the benthic isotope stack LR04 (Lisiecki and Raymo, 2005), a minimum age of 1.2 Ma has been proposed for the permanent lake phase of Lake Ohrid (Wagner et al., 2014). This minimum age is supported by the results from more detailed studies of the uppermost 247.8 mcd of the DEEP site sequence, which cover the last 637 kyr, according to an age model derived from tephrochronology and tuning of bio-geochemical proxy data to orbital parameters (Francke et al., 2016). The high-resolution data allow a better understanding of proxy variation over time and show that high TIC characterises interglacial periods and very low TIC represents glacial periods, as previously inferred from studies on core catcher material (Wagner et al., 2014). Indeed, a prominent TIC maximum at ~368 m blf in the core catcher samples from the DEEP site was presumed to represent the Marine Isotope Stage (MIS) 31 at 1.081-1.062 Ma (Wagner et al., 2014), which is regarded as one of the warmest interglacials at the onset of the Mid Pleistocene Transition (MPT; e.g., Melles et al., 2012). The lithology of the DEEP site sediment sequence indicates that lacustrine, hemi-pelagic sediments comprise the upper ~430 m blf, whereas littoral and fluvial sediments dominate below (Wagner et al., 2014). The transition from fluvial or littoral facies to hemi-pelagic sediments most likely indicates the onset of full lacustrine conditions in Lake Ohrid. Five TIC maxima below the presumed MIS 31 maximum and above the fluvial or littoral facies (cf., Wagner et al., 2014) could represent five additional interglacials, which would place the onset of hemi-pelagic sedimentation within MIS 41 and refines the minimum age of Lake Ohrid to ca. 1.3 Ma.

An age estimation for the onset of lacustrine sedimentation in the Lake Ohrid basin has been derived from comparing seismic and chronological information from core Co1202 recovered





in the north-eastern part of the lake (Fig. 1). Tracking seismic reflectors from this coring
location (~2 km from the DEEP site) to the central part of the lake allowed for the transfer of
chronological information of the core into the basin centre (Lindhorst et al., 2015). In
addition, the strength of the reflectors was correlated with chronological information and
glacial/interglacial cycles derived from pollen analyses at Lake Ioannina, 200 km to the South
of Lake Ohrid. Based on this information, an average sedimentation rate of 0.43 mm yr$^{-1}$ was
calculated for the last 450 kyr in the basin centre (Lindhorst et al., 2015). Using this
sedimentation rate for the maximum sediment fill of ~800 m blf observed in the basin centre,
resulted in an age of 1.9 Ma for the onset of sedimentation (Lindhorst et al., 2015). At the
DEEP site a somewhat lower average sedimentation rate of 0.39 mm yr$^{-1}$ can be calculated for
the upper 247.8 mcd or for the last 637 kyr (Francke et al., 2016). Sediment compaction with
increasing sediment depth (cf. Baumgarten et al., 2015) may have caused further lowering of
the calculated sediment accumulation rate downward and also would lead to older ages
compared to those based on a constant sedimentation rate of 0.43 mm yr$^{-1}$. However,
lacustrine, hemi-pelagic sediments only form the upper ~430 m blf of sediments at the DEEP
site, which represents only half of the maximum sediment fill equivalent to ~800 m blf. As
the underlying littoral and fluvial sediments most likely have significantly higher
sedimentation rates, the extrapolated age of 1.9 Ma for the onset of hemi-pelagic
sedimentation can be regarded as a tentative maximum age, assuming there were no major
phases of erosion and/or non-deposition.
Overall, based on this new geological information, the minimum and maximum age of Lake
Ohrid can be restricted to ca. 1.3 and 1.9 Ma, respectively. More precise age estimation will
be obtained by ongoing tephrostratigraphic work and paleomagnetic analyses, which may
reveal the existence of major reversals in the Earth's magnetic field, such as the Jaramillo
(1.075-0.991 Ma), Cobb Mountain (1.1938-1.1858 Ma), or Olduvai (1.968-1.781 Ma)
subchrons (Nowaczyk et al., 2013 and references therein).
These estimates of 1.3-1.9 Ma correspond well to evolutionary data for endemic Lake Ohrid
species obtained prior to the drilling campaign. Based on genetic information from extant
endemic species and molecular-clock analyses, the onset of intralacustrine speciation in
various groups of Lake Ohrid endemics (= "ancient lake species flocks") started between 1.4





Ma for the limpet genus *Acroloxus* (Albrecht et al., 2006) and 2.0 Ma for the endemic *Salmo*
*trutta* trout complex (Sušnik et al., 2006) and the *Dina* leech flock (Trajanovski et al., 2010).
Assuming that the origin of Lake Ohrid predates the onset of intralacustrine speciation events,
the latter authors suggested that the minimum age of Lake Ohrid is approximately 2.0 Ma.
However, they were not able to explain why the species flocks investigated differed in their
time of origination and why some of the flocks were as young as 1.3 Ma. A potential
explanation is now provided by the initial results of the SCOPSCO deep drilling campaign,
which indicate that persisting lacustrine conditions with pelagic or hemi-pelagic
sedimentation established between 1.9 Ma and 1.3 Ma ago. The period of lake establishment
and persisting lacustrine conditions may have comprised up to several hundred thousand
years, which in turn might have given rise to most species flocks in Lake Ohrid.

### 4.1.2   Origin

There is a broad consensus that the 40 km long and N-S-trending Ohrid graben basin
developed as part of the Alpine orogeny during a transtensional phase in the Late Miocene,
followed by an extensional phase since the Pliocene (e.g., Cvijić 1911; Aliaj et al., 2001;
Dumurdzanov et al., 2004; Reicherter et al., 2011; Lindhorst et al., 2015). There is little
consensus on the limnological origin of the lake itself, however. Albrecht and Wilke (2008)
summarized four related hypotheses. Three of these hypotheses favour an origin as part of a
marine ingression or a brackish-water lake system during the Miocene: the Mesohellenic
Trough hypothesis, the Tethys hypothesis, and the Lake Pannon hypothesis. A fourth
hypothesis favours a *de novo* origin, i.e., that Lake Ohrid formed in a dry polje fed by springs
during the Pliocene or Pleistocene. The latter is supported, in part, by the known existence of
substantial active karst aquifers (Matzinger et al., 2006) and the seismic data, which indicate
that Lake Ohrid formed in a relatively narrow and elongated valley (Lindhorst et al., 2015).
Moreover, sediments at the base of the DEEP site sequence are formed by gravel, which is
overlain by alternating peat layers, sand horizons, and fine-grained sediments, and contain a
relatively shallow, obligate fresh water diatom flora (Wagner et al., 2014). These sediments
indicate very dynamic environments, ranging from fluvial to slack water conditions, with
varying shallow water conditions, and support, in combination with the presumed Pleistocene
age of Lake Ohrid, the *de novo* hypothesis of lake formation.





### 4.2 Sediment architecture and basin development

In addition to information on the formation of the Ohrid basin, the hydro-acoustic data sets from Lake Ohrid can also provide knowledge on mass transport deposits (MTDs) and on long-term lake level change.

The evaluation of the seismic and hydro-acoustic data sets indicated that MTDs are only observed during the last ca. 340 ka in Lake Ohrid (Lindhorst et al., 2016). Older MTDs are not covered by the seismic profiles or may be masked by multiple reflections below 250-300 m sediment depth in the central part of the basin. Five major MTDs are detected during MIS 9, 7, and 6. Since ca. 80 ka, the number of MTDs increased, however this is accompanied by a trend of decreasing MTD volume. Due to the restricted vertical resolution of the seismic data sets, the age control of the MTDs is relatively imprecise. Nevertheless, it seems that the occurrence of MTDs is not driven by or a response to glacial/interglacial cyclicity, as they occur during glacials, interglacials, and their respective transitions. Although MTDs are detected throughout the entire basin (Lindhorst et al., 2016), they cluster along the major faults in the southeastern and northwestern part of the basin and are probably the result of fault activity and major earthquakes (Lindhorst et al., 2012; Wagner et al., 2012). No indications for these MTDs are found in the drill cores of the DEEP site. Hence, MTDs in the Ohrid basin apparently have a rather limited spatial extent and are not accompanied by basin-wide suspension clouds or turbidites. MTDs with a maximum thickness of <3 cm are observed in the DEEP site record, with clusters in MIS 8, late MIS 6, and MIS 2 (Francke et al., 2016). The thickness of these MTDs is significantly below the vertical resolution of the seismic data.

The hydro-acoustic data can also provide information about the tectonic history of the basin with respect to lake-level fluctuations. The minimum water depth can be estimated from measuring the depth difference of individual reflectors between their largest depth in the basins and the minimal depth of occurrence at the lake margins. The minimal depth of





occurrence for individual reflectors maybe a real reflection termination but in most cases,
individual reflectors cannot be traced further up because the shallowest areas of the lake basin
are not covered by the seismic and hydro-acoustic survey or reflectors could not be traced to
the shallower parts due to faults (Fig. 2). In a second step, linking these reflectors to the
chronological information from the DEEP site provides chronological information for the
minimum water depth. Tracing a reflector from ~275 m blf at the DEEP site, i.e., a reflector
located below the existing age model, supposes a minimum water depth of 300 m (Fig. 2).
Reflectors at the MIS 16/15 (~240 m blf) and the MIS13/12 boundaries (~190 m blf) suggest
minimum water depths of 300 m as well, thus exceeding the present day water depth of 293 m
(Fig. 2). The minimum water depth was reduced to 225 m at the MIS 9/8 boundary (~140 m
blf), to 200 m during MIS 8 (~100 m blf), and to 175 m during MIS 5 (47 m blf). In MIS 4
(20 m blf), the minimum water depth increased to 250 m, returning to a level similar to that
observed in the lower half of the record. Note that this method for estimating water depth
contains several sources of uncertainties. The actual water depth during each period may have
been much higher, as individual reflectors may continue to shallower water depths or even
above the present lake level but cannot be mapped due to missing data coverage in shallow
water depth, or reflectors may have been eroded during a following period of a lower lake
level. Ongoing subsidence might also have affected the shape of the individual reflectors and
potentially increased the maximum depth difference of individual reflectors. Nonetheless, the
data suggest a general trend from deeper waters from prior to MIS 16 through to MIS 13/12,
followed by decreasing water depths with a minimum in MIS 5 and a subsequent deepening
to present day lake level, which is in a broad agreement with humidity trends based on the
regional vegetation cover (Sadori et al., 2016). As a result, the deepening of the Lake Ohrid
basin was apparently not a continuous and gradational process; we assume that short or mid-
term changes reflect changes in water budgets while subsidence is a much slower process.
However, already at or shortly after the end of the MPT, the lake showed similar or even
higher water depths compared to present lake level. The seismic data do indicate periods of
very low lake levels or even a dry lake.
Mapping of the hydro-acoustic reflectors indicates that the shape of the Ohrid basin slightly
altered over time. Based on the isopleths, the deeper part of the basin changed from a more
elongated shape to a roundish shape during the last ca. 700 kyr, with a formation of a





secondary basin in the northwestern part of the lake after the MIS 13/12 boundary at 478 ka
(Fig. 2). This also reflects the extension of the lake basin.
**4.3 Tephrostratigraphic and environmental history**
4.3.1 Tephrostratigraphy
The DEEP site sequence drilled in 2013 provides the most complete tephrostratigraphic
record obtained from Lake Ohrid. A total of 39 tephra layers have been identified in the upper
247.8 mcd so far (Fig. 3; Leicher et al., 2016 and unpublished data). Major element analyses
(SEM-EDS/WDS; see Leicher et al., 2016 for details) on juvenile glass fragments suggest an
origin exclusively from Italian volcanic provinces. Of these tephra layers (OH-DP-0027 to
OH-DP-2060), 13 could be identified and correlated with known and dated widespread
eruptions (Leicher et al., 2016 and references therein). They include the Mercato tephra (OH-
DP-0027, 8.43–8.63 cal ka BP) from Somma-Vesuvius, the Y-3 (OH-DP-0115, 26.68–29.42
cal ka BP), the Campanian Ignimbrite/Y-5 (OH-DP-0169, 39.6 ± 1.6 ka), and the X-6 (OH-
DP-0404, 109 ± 2 ka) from the Campanian volcanoes, the P-11 (OH-DP-0499, 129 ± 6 ka)
from Pantelleria, the Vico B (OH-DP-0617, 162 ± 6 ka) from the Vico volcano, the Pozzolane
Rosse (OH-DP-1817, 457 ± 2 ka) and the Tufo di Bagni Albule (OH-DP-2060, 527 ± 2 ka)
from the Colli Albani volcanic district, and the Fall A (OH-DP-2010, 496±3 ka) from the
Sabatini volcanic field. Furthermore, a comparison of the Ohrid record with
tephrostratigraphic records of mid-distal Mediterranean archives enabled the identification of
less well-known tephra layers, such as the TM24-a/POP2 (OH-DP-0404, 101.8 ka; Regattieri
et al., 2015) from Lago Grande di Monticchio and the Sulmona basin, the SC5 (OH-DP-1955,
493.1 ± 10.9 ka) from the Mercure basin, and the A11/12 (OH-DP-2017, 511 ± 6 ka) from the
Acerno basin, whose specific volcanic sources are still poorly constrained. OH-DP-0624 was
tentatively correlated to the CF-V5/PRAD3225 layers from the Campo Felice basin/Adriatic
Sea and, thus to the Pitigliano Tuff from the Vulsini volcanic field (ca. 163 ka; Leicher et al.,
2016). However, recent tephrochronological results including $^{40}Ar/^{39}Ar$ of a tephra from the
Fucino Basin, central Italy, suggest that these tephras correspond to an un-known eruption
from the Neapolitan volcanic area at 158.8 ± 3.0 ka (Giaccio et al., 2016). In order to obtain a
consistent set of ages all $^{40}Ar/^{39}Ar$ were calculated by using the same flux standard (1.194 Ma





for ACs, which corresponds to FCs at 28.02 Ma). The chronological information of 11 of the
well-identified tephras from Lake Ohrid was used as 1st order tie points for the age-depth
model of the composite core, and complemented by tuning of sediment proxies to orbital
parameters, such as summer insolation and winter season length (Francke et al., 2016).
Fifteen additional tephra horizons have been identified within the lower hemi-pelagic section
of the DEEP site sequence between 248 and 450 mcd (Fig. 3) and are the subject of on-going
work. Although knowledge of tephrostratigraphy for the period >637 ka is restricted, a
combination of tephrochronological with paleomagnetic information should provide a robust
chronology for this part of the sequence.
With a total of at least 54 tephra layers intercalated in a continuous sediment succession of >
1.3 Ma, the tephrostratigraphic record from Lake Ohrid is a strong candidate to become the
template for central Mediterranean tephrostratigraphy, especially for the poorly-known and
explored Lower and Middle Pleistocene period. The tephrostratigraphic record may also help
to allow re-evaluation and improvement of the chronology of dated and undated tephra layers
from other key sites, such as the age of the Fall A tephra (Leicher et al., 2016). Moreover, the
tephras constitute valuable, independent tie points that resolve leads and lags between
changes in different components of the climate system and allow a synchronisation of the
Lake Ohrid record with other regional records (Zanchetta et al., 2016).
### 4.3.2 Environmental history
The examination of the environmental history of Lake Ohrid over the last 637 kyr focuses
both on long-term changes over several glacial/interglacial periods, and short-term changes on
the sub-orbital scale.
Long-term changes



The study of the long-term environmental history of Lake Ohrid and its surrounding area
includes the reconstruction of minimum lake levels based on hydro-acoustic information, by
vegetation changes in the catchment, and by internal lake proxies. According to the
established age model (Francke et al., 2016), hydro-acoustic (Lindhorst et al., 2015) and
borehole logging data (Baumgarten et al., 2015), the sediments deposited at 637 ka are now
located ~240 m blf at the DEEP site. If the altitude of the Lake Ohrid outlet or the bedrock
gap used by the river Crni Drim would have been the same as it is today (693.5 m a.s.l.), the
water depth of Lake Ohrid at 637 ka would have been more than 480 m. There is no evidence
in the seismic or sedimentological data for such a great water depth at that time, which
implies that subsidence or other tectonic activity affected the sediment succession in the lake
basin or the altitude of the outlet. Nevertheless, the hydro-acoustic data suggest a fairly deep
lake at the end of the MPT, with a water depth similar or even deeper than today (Figs 2 and
4). Shallower minimum water depths are tentatively indicated between MIS 9 and MIS 3,
with an absolute minimum during MIS 6 or MIS 5. Tectonic activity and the relative altitude
of the outlet are probably the most significant contributors to water depth variations in Lake
Ohrid. However, a comparison of the minimum water depth data with pollen data suggests
that climate change may also have triggered water-depth fluctuations. Although the Lake
Ohrid watershed was a refugial area for both temperate and montane trees during the glacial
periods of the last 500 kyr, the earlier glacials MIS 12, MIS 10, and MIS 8 were characterized
by grassland (Sadori et al., 2016). Such vegetation would require relatively humid conditions,
whereas steppe vegetation with *Artemisia* and pioneer taxa typical of dry conditions
dominated during MIS 6, MIS 4, and MIS 2 (Fig. 4; Sadori et al., 2016). Mesophilous
communities representing a Mediterranean-type climate are found in MIS 5 and the Holocene.
The overall progressive change from cooler and wetter conditions recorded during both
interglacial and glacial periods prior to 288 ka to subsequently warmer and drier interglacials
and glacials (Sadori et al., 2016) is consistent with the generally shallower minimum water
levels reconstructed by tracing hydro-acoustic reflectors throughout the basin. Moreover,
driest conditions and a maximum in steppe vegetation between 160-129 ka (Sadori et al.,
2016) correspond to a prominent lake-level lowstand and the formation of a subaquatic
terrace ~60 m below the present lake level in the northeastern Ohrid basin (Fig. 4; Lindhorst
et al., 2010). This lowstand was reconstructed based on hydro-acoustic studies and
tephrochronological information from two short sediment cores. Two tephras deposited on the
terrace were previously correlated with MIS 5 tephras C-20 (ca. 80 ka) and X5 (105 ± 2 ka)



(Sulpizio et al., 2010), and it was supposed that the formation of this terrace took place during
MIS 6 (Lindhorst et al., 2010). However, new tephrostratigraphic results suggest that the two
tephras instead correspond with Vico B (OH-DP-0617, 162 ± 6 ka) and CF-V5/PRAD3225
(OH-DP-0624, ca. 163 ka; Leicher et al., 2016). This constrains the formation of this terrace
to the earlier part of MIS 6 and the subsequent lake-level increase to late MIS 6 or early MIS
5, with a secondary lowstand around 100 ka (Fig. 4), which approximately follows the overall
trend of the minimum lake-level reconstruction.
Internal lake proxies confirm the long-term trend seen in pollen from generally wetter and
cooler interglacial and glacial periods between 637 ka and ca. 300 ka to drier and warmer
stages between 300 ka and the Present. The oxygen isotope composition of lake water
($\delta^{18}O_{lakewater}$), calculated from $\delta^{18}O$ of endogenic calcite, shows only moderate variability
between interglacial periods with a relatively stable climate from MIS 15 to MIS 13,
progressively wetter conditions during MIS 11 and MIS 9, and increasingly evaporated, drier
conditions in more recent interglacials (Fig. 4; Lacey et al., 2016). In particular, higher
$\delta^{18}O_{lakewater}$ through MIS 5 and the Holocene indicate higher evaporation due to dry and warm
conditions prevailing under a Mediterranean-type climate. During glacials calcite is typically
absent, however $\delta^{18}O_{lakewater}$ reconstructed from early diagenetic siderite shows a more
pronounced long-term shift, with values being consistent with the adjacent interglacials
during MIS 14, MIS 12, and MIS 10, a transition to lower values through MIS 8, and very
low $\delta^{18}O_{lakewater}$ during MIS 6, MIS 4, and MIS 2 (Fig. 4). The similarity between interglacial
and glacial lake water prior to ca. 300 ka suggests that Lake Ohrid may have experienced
regular and complete mixing, as calcite and siderite form in different environments; calcite in
surface waters during summer months and siderite as a product of early diagenesis in the
surface sediments. Lower average $\delta^{18}O_{lakewater}$ before ca. 300 ka indicates moderate summer
temperatures (reduced seasonality). It may also suggest higher activity of the karst system due
to more precipitation and/or a higher lake level of neighbouring Lake Prespa. Subsequently, a
trend to higher $\delta^{18}O_{lakewater}$ during interglacials indicates stronger rates of summer evaporation
and drier conditions, and lower $\delta^{18}O_{lakewater}$ in glacial periods suggests isotopically fresh
conditions most likely due to low evaporation and a higher influence of winter precipitation
(increased seasonality), which supports the interpretation of the palynological record.





Increasing summer aridity towards present is also backed by the gradual increase of
Mediterranean taxa pollen percentages.
A transition from generally wetter and cooler to drier and warmer conditions is also indicated
by a shift from relatively invariant and low TOC prior to ca. 300 ka towards more fluctuating
and higher TOC, particularly during the more recent interglacials (Fig. 4; Francke et al.,
2016). Wetter and cooler conditions after the MPT drive a high activity of the karst system
and intense mixing of the water column, thus promoting decomposition of organic matter.
This would, in turn, increase the supply of sulphur to the sediments and allow for the
formation of greigite (Fig. 4; Just et al., 2016). A greater activity of the karst system and
associated high ion ($Ca^{2+}$, $HCO_3^-$) input is further supported by the relatively high TIC during
MIS 15, MIS 14, and MIS 13 (Fig. 4; Francke et al., 2016). Pollen data suggest moderate
summer temperatures, i.e., conditions that would have favoured mixing and, hence, increased
organic matter degradation. Conversely, drier and warmer conditions after ca. 300 ka likely
reduced mixing of the water column during the interglacials, which would lead to anoxic
bottom waters and a better preservation of organic matter. Such conditions are indicated by
the predominant formation of siderite during these more recent glacials, when limited sulphur
content of sediments may have prevented the formation of greigite (Fig. 4), with siderite
precipitating instead.
The maximum sedimentation rate during early MIS 6 (Francke et al., 2016) correlates well
with the formation of the subaquatic terrace located at 60 m below the present lake level (Fig.
4; Lindhorst et al., 2010). The lower lake level led to exposure and erosion of formerly
shallow parts of the lake and a lower distance from inlets to the central part of the lake.
However, there is no indication, e.g., in isotope or redox sensitive data, for an endorheic lake
at that time or any other time during the last 637 kyr. It thus seems that the outlet was always
active and climate driven lake-level change may have been compensated at least partly by
tectonic activity.
Sub-orbital changes





On a sub-orbital scale, prominent environmental changes in the Northern Hemisphere that
potentially affected Lake Ohrid include Dansgaard-Oeschger (D/O) and Heinrich events (HE).
D/O events are a pervasive feature of the last glacial (e.g., Wolff et al., 2010) and also of
older glacial periods (Stein et al., 2009; Naafs et al., 2014). They are likely related to
variations in the Atlantic Meridional Overturning Circulation (AMOC) and are recorded as
climatic perturbations in many marine and terrestrial records (e.g., Genty et al., 2002; Rohling
et al., 2003; Naafs et al., 2014; Seierstad et al., 2014; Stockhecke et al., 2016). In the eastern
Mediterranean, D/O events may have influenced regional hydrology and led to large-scale
droughts during the past four glacial cycles (Stockhecke et al., 2016). HE are distinctively
represented by deposition of ice rafted debris (IRD) in North Atlantic marine cores (e.g.,
Hemming et al., 2004), and are also well documented to have had an imprint on marine and
terrestrial records for the last glacial and beyond (e.g., Sanchez-Goni et al., 2002; Martrat et
al., 2004; Naafs et al., 2013). At the IODP drill site U1308 in the North Atlantic, HE are first
indicated during MIS 16 and are represented by ice-rafted debris (IRD) layers that are rich in
detrital carbonate and poor in biogenic carbonate (Hodell et al., 2008). It has been speculated
that ice volume and the duration of glacial conditions surpassed a critical threshold during
MIS 16 and activated the dynamic processes responsible for Laurentide Ice Sheet instability
in the region of Hudson Strait, which led to increased iceberg discharge and weakening of
thermohaline circulation in the North Atlantic (Hodell et al., 2008).
MIS 12 is considered to be one of the most severe glacials during the Quaternary, with the
lowest summer sea surface temperatures (SST) recorded across multiple records (e.g.,
Shackleton 1987; Naafs et al., 2013, 2014; Rohling et al., 2014). Abrupt sea surface warming
events of 3-6°C in the mid-latitude North Atlantic during MIS 12 likely reflect the imprint of
D/O events and probably had a substantial impact on global climate (Naafs et al., 2014). In
contrast to the observations from MIS 16, a temporal lag between the occurrence of IRD and
surface water cooling during MIS 12 implies that HE were not the cause for a weakening of
the thermohaline circulation in the North Atlantic at this time (Naafs et al., 2014).
High-resolution records from the Mediterranean region, which can be used to test a larger
regional or even global impact of D/O and HE during MIS 16 or MIS 12, are scarce (e.g.,
Hughes et al., 2006; Tzedakis et al., 2006; Girone et al., 2013; Capotondi et al., 2016). A





multi-proxy record with lithological, geochemical, and isotope data from the Sulmona basin
in central Italy covering MIS 12 shows pronounced hydrological variability at orbital and
millennial time scales, which replicates North Atlantic and western Mediterranean SST
fluctuations (Fig. 5; Regattieri et al., 2016). Several short-term fluctuations in the MIS 12
Sulmona record most likely reflect sub-orbital scale hydrological variations, and are likely
related to reduced precipitation sourcing from the North Atlantic due to episodes of iceberg
melting, and IRD deposition at the west Iberian margin (Regattieri et al., 2016 and references
therein). However, as the timing of these IRD events at the western Iberian margin was used
to improve the chronology of the Sulmona record, the correlation of hydrological variations in
central Italy and IRD deposition in the North Atlantic is not fully independent.
At Lake Ohrid and further to the East, the arboreal pollen concentration in the Tenaghi
Philippon record from Greece correlates well with the general pattern of the sea surface
temperatures in the North Atlantic during MIS 12 (Fig. 5; Tzedakis et al., 2006). The
resolution of the existing record is too low yet to allow a clear identification of D/O or HE
related climate change. The high-resolution record from Lake Van in eastern Turkey also
cannot be used for testing the climatic impact of D/O or HE on the eastern Mediterranean, as
the sediments of MIS 12 and the onset of MIS 11 are disturbed and lack independent age
control (Stockhecke et al., 2014).
The new high-resolution record from the DEEP site in Lake Ohrid now offers the possibility
to assess the impact of D/O or HE during MIS 12 on a broader regional scale, particularly as
it provides two absolute tephra age control points with ages centred at 493.1 ± 10.9 and 457 ±
2 ka (Fig. 5; Francke et al., 2016; Leicher et al., 2016). During MIS 12, potassium (K) shows
a long-term increase, which supports the overall trend towards colder temperatures, such as
can be inferred from other marine, terrestrial, or synthetic climate records (Fig. 5). K
represents the proportion of clastic, terrigenous matter relative to the content of carbonate
(reflected by TIC) and organic matter (reflected by TOC and bSi). TOC was used to infer the
severity of glacials at Lake Ohrid (Francke et al., 2016) and shows a remarkable saw tooth
pattern during MIS 12, which resembles fluctuations in SST related to D/O variability from
the North Atlantic marine record U1313 (Fig. 5; Naafs et al., 2014). Higher TOC is favoured
by both increased overall productivity (on land and in the water column) as well as increased




organic matter preservation, with the latter resulting from oxygen depletion of the bottom
water due to enhanced thermal stratification, decreased mixing, and higher temperatures.
These higher temperatures at Lake Ohrid likely correlate with higher SST in the North
Atlantic. The TOC record from Lake Ohrid thus would be the first terrestrial record to
indicate D/O cycle-related teleconnections between the North Atlantic thermohaline
circulation and the climate in the northeastern Mediterranean region during MIS 12.
Interestingly, the dominant *Pinus* pollen abundance in the vegetation record indicates a
regular ~8 kyr variability during MIS 12 and 10), for which high-resolution analysis is now
being performed (Figure 2 in Sadori et al., 2016).
The environmental impact of HE or other short-term climate events has been studied in detail
for the last glacial cycle in records from lakes Ohrid and Prespa. Based on pollen and diatom
analyses, HE in the North Atlantic during MIS 4 to MIS 2 led to short spells of very dry and
cold conditions superimposed on the glacial conditions (Panagiotopoulos et al., 2014;
Cvetkoska et al., 2015). Moreover, there is an increased formation of Fe and Mn concretions
in Lake Prespa sediments, most likely driven by a significant shift in the bottom water redox
conditions (Wagner et al., 2010). According to diatom studies spanning the last 92 ka, Lake
Prespa experienced significant regime shifts that are correlated with lake level fluctuations
and changes between (oligo-) meso- and eutrophic conditions (Cvetkoska et al., 2016). Lake
Ohrid seems to be less sensitive to short-term climate change due to its higher volume to
surface area ratio (e.g., Wagner et al., 2010; Leng et al., 2013). It does not indicate sub-orbital
time scale lake-level changes and shifted between ultra oligo- and oligotrophic conditions
during the last 92 kyr (Cvetkoska et al., 2016). However, the formation of Fe and Mn
concretions and the occurrence of siderite indicate that Lake Ohrid is also sensitive to shifts in
the bottom water redox conditions (Lacey et al., 2016). During MIS 12, Fe peaks in XRF data
are positively correlated with TIC and indicate the formation of early-diagenetic siderite in
response to a shift in bottom water redox conditions towards a more oxic environment (Fig. 5;
Francke et al., 2016; Lacey et al., 2016). The Fe peaks during the coldest period of this glacial
match particularly well with the number of IRD grains and with maxima in the quartzite- or
dolomite-calcite ratio in the U1313 record from the North Atlantic (Fig. 5). The latter are
interpreted as millennial ice-rafting driven events (Voelker et al., 2010; Naafs et al., 2011,



2013) and thus demonstrate that North Atlantic HE may have caused changes in internal lake
conditions, such as bottom water redox conditions.
One of the HE, the H4 event at 40.4–38.4 ka, is superimposed by another short-term event,
the eruption from the Campi Flegrei volcanoes 39.6 ± 1.6 ka. This eruption is one of the most
severe volcanic eruptions during the Pleistocene and left a 15 cm thick tephra known as
Campanian Ignimbrite or Y-5 marine tephra layer in the records from lakes Ohrid and Prespa
(e.g., Wagner et al., 2009; Vogel et al., 2010b; Damaschke et al., 2013). High-resolution
studies of diatoms in both lake sediment records indicated little evidence for a response of the
diatom community related to the H4 event, but a clear and rapid change following tephra
deposition (Jovanovska et al., 2016). This strong change is likely due to fertilisation and the
availability of nutrients, particularly silica, such as it was shown in laboratory studies and
leaching experiments of tephra with Lake Ohrid water (D´Addabbo et al., 2015). After the
initial response, diatom community compositions in lakes Ohrid and Prespa returned to their
quasi pre-disturbance state. In Lake Ohrid, the recovery time was ca. 1100 years vs. ca. 4000
years in Lake Prespa (Jovanovska et al., 2016). Although both lakes are resilient to short-term
environmental change, it seems that Lake Ohrid is even more resilient than Lake Prespa,
likely due to differences in geology, lake age, limnology, and intrinsic parameters of the
diatom proxies (Jovanovska et al., 2016).
**4.4  Drivers of biodiversity change**
One of the major interdisciplinary goals of the SCOPSCO project is to infer the drivers of the
extraordinary endemic biodiversity in Lake Ohrid, in general, and to evaluate the influence of
major environmental events on evolutionary processes, in particular. Lake Ohrid thus serves
as a model system to address questions that have puzzled evolutionary biologists for decades.
These questions include the problem whether the high number of endemic species is mainly a
result of an accumulation of relic species ('reservoir function') and/or of a high rate of
intralacustrine speciation ('cradle function'). Moreover, if intralacustrine speciation plays a
significant role, is it primarily driven by geographic or environmental gradients during
periods of relatively constant environmental conditions, possibly supported by a high



ecosystem resilience of the lake, or does ongoing environmental change lead to an increase
(or decrease) in rates of species diversification? Finally, what role do potentially
'catastrophic' environmental fluctuations play, such as lake level change or significant
changes in the trophic state?

### 4.4.1 Reservoir vs. cradle function of Lake Ohrid

As discussed in Föller et al. (2015), ancient lakes have often been considered to serve as
evolutionary or geographic refugia, either harboring old and distinct lineages or enabling the
accumulation of species from extralimital areas during periods of adverse environmental
changes, respectively ('reservoir function'). However, previous evolutionary studies in Lake
Ohrid on selected animal taxa could not demonstrate the existence of such relict species
(sensu Grandcolas et al., 2014), either because ancestral distribution ranges are largely
unknown (e.g., Schultheiß et al., 2008) or the native species are not extraordinarily old (e.g.,
Albrecht et al., 2008; Hauswald et al., 2008). Instead, intralacustrine speciation after
immigration events prevails. Most endemic animal species in Lake Ohrid are considerably
younger than the lake itself and form a monophyletic clade (also see Section 4.1.2.). This
suggests that the high endemic species richness in Lake Ohrid invertebrates is predominantly
a result of intralacustrine diversification ('cradle function', e.g., Albrecht et al., 2006, 2008;
Wilke et al., 2007; Schultheiß et al., 2008; Wysocka et al., 2014; Föller et al., 2015).
Interestingly, the situation is different for plant species inhabiting the surrounding of Lake
Ohrid. For example, the existing pollen record from the DEEP site sequence, which covers
the last 500 kyr, indicates that the Lake Ohrid catchment has indeed been a refugial area for
both temperate and montane trees during glacial periods (Sadori et al., 2016).

### 4.4.2 Impact of environmental change on species diversification

Ancient lakes are often considered to be comparatively stable systems, potentially resulting in
constant diversification rates (i.e., speciation minus extinction rates) over time. Nonetheless,
several factors, often related to environmental, geological, or climatic changes, and depending
on the genetic features of the species, have been suggested to affect the tempo of





diversification in ancient lake species flocks. Accordingly, phases of rapid environmental
fluctuations may lead to net evolutionary change. Diversification rates may be higher in the
initial phase of lake colonisation and may decline once niche space is increasingly occupied.
Alternately, there might be a pronounced lag phase between the colonization of a lake and the
onset of subsequent diversification (reviewed in Föller et al., 2015).
Although high-resolution sediment-core analyses, covering the last 637 kyr, indicate that
Lake Ohrid experienced several environmental changes, phylogenetic studies on a
microgastropod group using lineage-through-time plots and diversification-rate analyses did
not reveal significant changes in this rate over time (Föller et al., 2015). Moreover, diatom
community analyses conducted from the DEEP sediment cores could not show extinction
events due to major environmental events such as tephra deposition (Jovanovska et al., 2016;
for details see section 4.3.2) and climate change over the last 92 kyr (Cvetkoska et al., 2016).
However, the potential for a regime shift increases with recent human impact on the diatom
flora of both lakes Ohrid (Zhang et al., 2016) and Prespa (Cvetkoska et al., 2015) although,
again, Ohrid appears to be more well-buffered from eutrophication than Prespa.
The reasons for the relatively constant diversification rate over time observed in
microgastropods and the lack of diatom extinction events during the Late
Pleistocene/Holocene remain largely unknown. However, a lack of environmental induced
extinction events in Lake Ohrid and/or a high resilience of its ecosystems may have played a
role (Föller et al., 2015; Cvetkoska et al., 2016; Jovanovska et al., 2016). Nonetheless, though
environmental changes may have had only a minor direct effect on diversification processes
in endemic taxa of Lake Ohrid, these changes potentially altered the abundance and
community compositions of diatoms and ostracods (e.g., Belmecheri et al., 2010; Reed et al.,
2010; Zhang et al., 2016), thus indirectly affecting speciation processes. In fact, the analysis
of the gastropod community in Lake Ohrid implied the presence of both geographical and
ecological speciation due to physical barriers and divergence across environmental or life
history gradients, respectively (Hauffe et al., 2016).



Another aspect of environmental change is the impact of anthropogenic activity on species
composition, diversity, and diversification. As previously suggested, Lake Ohrid is facing a
"creeping biodiversity crisis", as increasing human impact in and around the lake already
jeopardises endemic species (Kostoski et al., 2010). For example, the presence of globally
invasive species has been recently demonstrated for Lake Ohrid (Albrecht et al., 2014).
Moreover, human-mediated environmental change is also predicted to alter the trophic state
of the lake (e.g., Matzinger et al., 2006). Given the small size of both the lake and its
catchment, increasing negative effects on the endemic biodiversity of Lake Ohrid and the
respective habitats are foreseeable and will likely foster extirpation. Only concerted and
international conservation activities might help mitigating the human impact on the sensitive
and highly biodiverse ecosystem of Lake Ohrid.
**5   Conclusions and outlook**
The SCOPSCO deep drilling project was initiated in 2004 and aimed at inferring (i) the age
and origin of Lake Ohrid (Former Yugoslav Republic of Macedonia/Republic of Albania), (ii)
its regional seismotectonic history, (iii) volcanic activity and climate change in the central
northern Mediterranean region, and (iv) the drivers of biodiversity and endemism. The project
included phylogenetic and metacommunity analyses of living invertebrates and sampling
from main modern terrestrial organic matter pools from the lake and its surroundings, seismic
and hydro-acoustic surveys of the lake internal sediment architecture, and the recovery of
surface sediments and sediment cores. Within the framework of the International Continental
Scientific Drilling Program (ICDP) a deep drilling in Lake Ohrid took place in spring 2013
and provided a 584 m long sediment sequence from the central part (DEEP site) of the lake.
Initial results of the study of this sediment sequence in combination with the results of the
biological and geophysical as well as former sedimentological studies reveal that the Ohrid
basin formed during the Miocene and Pliocene. Lake Ohrid established between 1.9 and 1.3
Myr ago and provides a continuous record of distal tephra deposition and climatic and
environmental change in the central northern Mediterranean region. With its geographical
location, the Lake Ohrid record provides a unique opportunity to align marine records from
the North Atlantic with long-term and independently dated terrestrial archives in the Northern
and Eastern Mediterranean, such as the records from the Sulmona basin, Tenaghi Philippon,




Lake Van, or Dead Sea. This is a major precondition to disentangle longitudinal climate
gradients and investigate leads and lags circumventing age model uncertainties.
More detailed studies exist meanwhile on the upper 247.8 m of the DEEP site sediment
sequence and indicate that this part represents the last 637 kyr. Over this period, Lake Ohrid
experienced significant environmental change, which is related to orbital-scale climate forcing
and regional geological events. These changes apparently did not cause major extinction
events in Lake Ohrid, as evident from both the microgastropod phylogeny and the diatom
fossil record. The potential high resilience of the ecosystem to past climatic and
environmental changes together with relatively low extinction rates may explain the
extraordinary degree of endemic biodiversity in the lake. Ongoing biological studies and more
detailed analyses of the early stages of Lake Ohrid basin, based on the now accessible
sediment records, will help to better understand the drivers of biological diversification and
endemism. Lake Ohrid is thus a key site to further resolve the link between biological and
geological evolution and should centre our attention on protecting the endemic community
from a substantial biodiversity crisis due to the increasing anthropogenic impact.

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

Preliminary results from the study of flora and vegetation of Ohrid lake, Natura
Montenegrina, 9, 253-264, 2010.
Jovanovska, E., Cvetkoska, A., Hauffe, T., Levkov, Z., Wagner, B., Sulpizio, R., Francke, A.,
Albrecht, A., and Wilke, T.: Differential resilience of ancient sister lakes Ohrid and Prespa to
environmental disturbances during the Late Pleistocene, Biogeosciences, 13, 1149-1161,
21 2016.

Just, J., Nowaczyk, N., Sagnotti, L., Francke, A., Vogel, H., Lacey, J.H., and Wagner, B.:
Climatic control on the occurrence of high-coercivity magnetic minerals and preservation of
greigite in a 640 ka sediment sequence from Lake Ohrid (Balkans), Biogeosciences, 13, 1179-
26 1196, 2016.



Kostoski, G., Albrecht, C., Trajanovski, S., and Wilke, T.: A freshwater hotspot under
pressure – assessing threats and identifying conservation needs for ancient Lake Ohrid.
Biogeosciences, 7, 3999-4015, 2010.
Lacey, J., Francke, A., Leng, M. J., Vane, C. H., and Wagner, B.: A high resolution Late
Glacial to Holocene record of environmental change in the Mediterranean from Lake Ohrid
(Macedonia/Albania), Int. J. Earth Sci., 104, 1623-1638, 2015.
Lacey, J. H., Leng, M. J., Francke, A., Sloane, H. J., Milodowski, A., Vogel, H., Baumgarten,
H., and Wagner, B.: Mediterranean climate since the Middle Pleistocene: a 640 ka stable
isotope record from Lake Ohrid (Albania/Macedonia), Biogeosciences, 13, 1801-1820, 2016.
Leicher, N., Zanchetta, G., Sulpizio, R., Giaccio, B., Nomade, S., Wagner, B., and Francke,
A.: First tephrostratigraphic results of the DEEP site record in Lake Ohrid, Macedonia,
Biogeosciences, 13, 2151-2178, 2016.
Leng, M. J., Baneschi, I., Zanchetta, G,. Jex, C. N., Wagner, B., and Vogel, H.: Late
Quaternary palaeoenvironmental reconstruction from Lakes Ohrid and Prespa
(Macedonia/Albania border) using stable isotopes, Biogeosciences 7, 3109-3122, 2010.
Leng, M. J., Wagner, B., Aufgebauer, A., Panagiotopoulos, K., Vane, C., Snelling, A.,
Haidon, C., Woodley, E., Vogel, H., Zanchetta, G., Sulpizio, R., and Baneschi, I.:
Understanding past climatic and hydrological variability in the Mediterranean from Lake
Prespa sediment isotope and geochemical record over the last glacial cycle, Quaternary Sci.
Rev., 66, 123-136, 2013.
Lindhorst, K., Vogel, H., Krastel, S., Wagner, B., Hilgers, A., Zander, A., Schwenk, T.,
Wessels, M., and Daut, G.: Stratigraphic analysis of lake level fluctuations in Lake Ohrid: an





integration of high resolution hydro-acoustic data and sediment cores, Biogeosciences, 7,
3531–3548, 2010.
Lindhorst, K., Gruen, M., Krastel, S., and Schwenk, T.: Hydroacoustic Analysis of Mass
Wasting Deposits in Lake Ohrid (FYR Macedonia/Albania), in: Submarine Mass Movements
and Their Consequences, edited by: Yamada, Y., Kawamura, K., Ikehara, K., Ogawa, Y.,
Urgeles, R., Mosher, D., Chaytor, J., and Strasser, M., Springer, the Netherlands, 245–253,

8  2012.

Lindhorst, K., Krastel, S., Reicherter, K., Stipp, M., Wagner, B., and Schwenk, T.:
Sedimentary and tectonic evolution of Lake Ohrid (Macedonia/Albania), Basin Res., 27, 84–

12  101, 2015.

Lindhorst, K., Krastel, S., and Baumgarten, H.: Mass Wasting history within Lake Ohrid
Basin (Macedonia/Albania) over the last 600 ka, Submarine Mass Movements and their
Consequences: 7th International Symposium. G. Lamarche, J. Mountjoy, S. Bull et al. Cham,
Springer International Publishing: 291-300, 2016.
Lisiecki, L. E. and Raymo, M. E.: A Pliocene-Pleistocene stack of 57 globally distributed
benthic _18O records, Paleoceanography, 20, PA1003, 2005.
Matevski, V., Carni, A., Avramovski, O., Juvan, N., Kostadinovski, M., Košir, P., Marinšek,
A., Paušic, A., and Šilc, U.: Forest Vegetation of the Galicica Mountain Range in Macedonia,
Založba ZRC, Ljubljana, 2011.
Martrat, B., Grimalt, J. O., Lopez-Martinez, C., Cacho, I., Sierro, F. J., Flores, J. A., Zahn, R.,
Canals, M., Curtis, J. H., and Hodell, D. A.: Abrupt temperature changes in the Western
Mediterranean over the past 250,000 years, Science, 306, 1762-1765, 2004.




Matter, M., Anselmetti, F. S., Jordanoska, B., Wagner, B., Wessels, M., and Wüest, A.:
Carbonate sedimentation and effects of eutrophication observed at the Kališta subaquatic
springs in Lake Ohrid (Macedonia), Biogeosciences, 7, 3755–3767, 2010.
Matzinger, A., Spirkovski, Z., Patceva, S., and Wüest, A.: Sensitivity of ancient Lake Ohrid
to local anthropogenic impacts and global warming, J. Great Lakes Res., 32, 158–179, 2006.
Matzinger, A., Schmid, M., Veljanoska-Sarafiloska, E., Patceva, S., Guseska, D., Wagner, B.,
Müller, B., Sturm, M., and Wüest, A.: Eutrophication of ancient Lake Ohrid: Global warming
amplifies detrimental effects of increased nutrient inputs, Limnol. Oceanogr., 52, 338–353,

11   2007.

Naafs, B. D. A., Hefter, J., Ferretti, P., Stein, R., and Haug, G. H.: Sea surface temperatures
did not control the first occurrence of Hudson Strait Heinrich Events during MIS 16,
Paleoceanography 26, PA4201, 2011.
Naafs, B. D. A., Hefter, J., and Stein, R.: Millennial-scale ice rafting events and Hudson Strait
Heinrich(-like) events during the late Pliocene and Pleistocene: A review, Quaternary Sci.
Rev., 80, 1-28, 2013.
Naafs, B. D. A., Hefter, J., and Stein, R.: Dansgaard-Oeschger forcing of sea surface
temperature variability in the midlatitude North Atlantic between 500 and 400 ka (MIS 12),
Paleoceanography, 29, 1024-1030, doi:10.1002/ 2014PA002697, 2014.
National Research Council.: Freshwater ecosystems: Re-vitalizing educational programs in
limnology. National Academy Press, Washington, D.C. 364 p., 1996.
Nowaczyk, N. R., Haltia, E. M., Ulbricht, D., Wennrich, V., Sauerbrey, M. A., Rosén, P.,
Vogel, H., Francke, A., Meyer- Jacob, C., Andreev, A. A., and Lozhkin, A. V.: Chronology of

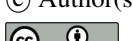


Lake El'gygytgyn sediments – a combined magnetostratigraphic, palaeoclimatic and orbital
tuning study based on multi-parameter analyses, Clim. Past, 9, 2413–2432, 2013.
Panagiotopoulos, K., Böhm, A., Leng, M. J., Wagner, B., and Schäbitz, F.: Climate variability
over the last 92 ka in SW Balkans from analysis of sediments from Lake Prespa, Clim. Past,
6  10, 643-660, 2014.
Popovska, C. and Bonacci, O.: Basic data on the hydrology of Lakes Ohrid and Prespa,
Hydrol. Proc., 21, 658–664, 2007.
Reed, J. M., Cvetkoska, A., Levkov, Z., Vogel, H., and Wagner, B.: The last glacial-
interglacial cycle in Lake Ohrid (Macedonia/Albania): testing diatom response to
climate, Biogeosciences, 7, 3083-3094, 2010.
Regattieri, E., Giaccio, B., Zanchetta, G., Drysdale, R. N., Galli, P., Nomade, S., Peronace, E.,
and Wulf S.: Hydrological variability over Apennine during the Early Last Glacial
precession minimum, as revealed by a stable isotope record from Sulmona basin, central Italy,
J. Quat. Sci., 30, 19-31, 2015.
Regattieri, E., Giaccio, B., Galli, P., Nomade, S., Peronace, E.,Messina P., Sposato, A.,
Boschi, C., and Gemelli, M.: A multi-proxy record of MIS 11-12 deglaciation and glacial
MIS 12 in-stability from the Sulmona Basin (central Italy), Quaternary Sci. Rev., 132, 129–
23  145, 2016.
Reicherter, K., Hoffmann, N., Lindhorst, K., Krastel, S., Fernandez-Steeger, T., Grützner, C.,
and Wiatr, T.: Active Basins and Neotectonics: Morphotectonics of the Lake Ohrid Basin
(FYROM and Albania), Z. Dtsch. Ges. Geowiss., 162, 217–234, 2011.





Rohling, E. J., Mayewski, P. A., and Challenor, P.: On the timing and mechanism of
millennial-scale climate variability during the last glacial cycle, Clim. Dynam., 20, 257–267,
3  2003.

Sadori, L., Koutsodendris, A., Panagiotopoulos, K., Masi, A., Bertini, A., Combourieu-
Nebout, N., Francke, A., Kouli, K., Joannin, S., Mercuri, A. M., Peyron, O., Torri, P.,
Wagner, B., Zanchetta, G., Sinopoli, G., and Donders, T. H.: Pollen-based
paleoenvironmental and paleoclimatic change at Lake Ohrid (SE Europe) during the past 500
ka, Biogeosciences, 13, 1423-1437, 2016.
Schneider, S., Cara, M., Eriksen, T. E., Budzakoska Goreska, B., Imeri, A., Kupe, L.,
Loshkoska, T., Patceva, S., Trajanovska, S., Trajanovski, S., Talevska, M., and Veljanovska
Sarafilovska, E.: Eutrophication impacts littoral biota in Lake Ohrid while water phosphorus
concentrations are low, Limnologica, 44, 90–97, 2014.
Schreiber, K., Hauffe, T., Albrecht, C., and Wilke, T.: The role of barriers and gradients in
differentiation processes of pyrgulinid microgastropods of Lake Ohrid, Hydrobiologia, 682,
61–73, 2012.
Schultheiß, R., Albrecht, C., Bößneck, U., and Wilke, T.: The neglected side of speciation in
ancient lakes: phylogeography of an inconspicuous mollusc taxon in lakes Ohrid and Prespa,
Hydrobiologia, 615, 141–156, 2008.
Seierstad, I. K., Abbott, P. M., Bigler, M., Blunier, T., Bourne, A.J., Brook, E., Buchardt, S.
L., Buizert, C., Clausen, H. B., Cook, E., Dahl-Jensen, D., Davies, S. M., Guillevic, M.,
Johnsen, S. J., Pedersen, D. S., Popp, T. J., Rasmussen, S. O., Severinghaus, J. P., Svensson,
A., and Vinther, B. M.: Consistently dated records from the Greenland GRIP, GISP2 and
NGRIP ice cores for the past 104 ka reveal regional millennial-scale d18O gradients with
possible Heinrich event imprint, Quaternary Sci. Rev., 106, 29-46, 2014.



Shackleton, N. J.: Oxygen isotopes, ice volume and sea level, Quaternary Sci. Rev., 6, 183-
2   190, 1987.

Stanković, S.: The Balkan Lake Ohrid and its living world, Dr. W. Junk, The Hague, 1960.
Stockhecke, M., Kwiecien, O., Vigliotti, L., Anselmetti, F. S., Beer, J., Çagatay, M. N.,
Channell, J. E. T., Kipfer, R., Lachner, J., Litt, T., Pickarski, N., and Sturm, M.:
Chronostratigraphy of the 600,000 year old continental record of Lake Van (Turkey),
Quaternary Sci. Rev., 104, 8–17, 2014.
Stockhecke, M., Timmermann, A., Kipfer, R., Haug, G. H., Kwiecien, O., Friedrich, T.,
Menviel, L., Litt, T., Pickarski, N., Anselmetti, F. S.: Millennial to orbital-scale variations of
drought intensity in the Eastern Mediterranean, Quaternary Sci. Rev., 133, 77-95, 2016.
Sulpizio, R., Zanchetta, G., D´Orazio, M. D., Vogel, H., and Wagner, B.: Tephrostratigraphy
and tephrochronology of the lakes Ohrid and Prespa, Balkans, Biogeosciences 7, 3273-3288,
17  2010.

Sušnik, S., Knizhin, I., Snoj, A., and Weiss, S.: Genetic and morphological characterization of
a Lake Ohrid endemic, *Salmo* (*Acantholingua*) *ohridanus* with a comparison to sympatric
*Salmo trutta*, J. Fish Biol., 68, Supplement A, 2–23, 2006.
Thienemann, A.: Untersuchungen über die Beziehung zwischen dem Sauerstoffgehalt des
Wassers und der Zusammensetzung der Fauna in norddeutschen Seen, A. Hydrobiol., 12, 1-
25  65, 1918.

Trajanovski, S., Albrecht, C., Schreiber, K., Schultheiß, R., Stadler, T., Benke, M., and Wilke,
T.: Testing the spatial and temporal framework of speciation in an ancient lake species flock:



the leech genus *Dina* (Hirudinea: Erpobdellidae) in Lake Ohrid, Biogeosciences, 7, 3387–

2  3402, 2010.

Tzedakis, P.C., Hooghiemstra, H., and Pälike, H.: The last 1.35 million years at Tenaghi
Philippon: revised chronostratigraphy and long-term vegetation trends, Quaternary Sci. Rev.,
25, 3416–3430, 2006.
Voelker, A. H. L., Rodrigues, T., Billups, K., Oppo, D. W., McManus, J. F., Stein, R., Hefter,
J., and Grimalt, J. O.: Variations in mid-latitude North Atlantic surface water properties
during the mid-Brunhes (MIS 9-14) and their implications for the thermohaline circulation,
Clim. Past, 6, 531-552, doi:10.5194/cp-6-531-2010, 2010.
Vogel, H., Wagner, B., Zanchetta, G., Sulpizio, R., and Rosén, P.: A paleoclimate record with
tephrochronological age control for the last glacial–interglacial cycle from Lake Ohrid,
Albania and Macedonia, J. Paleolimnol., 44, 295–310, 2010a.
Vogel, H., Zanchetta, G., Sulpizio, R., Wagner, B., and Nowaczyk, N.: A tephrostratigraphic
record for the last glacial–interglacial cycle from Lake Ohrid, Albania and Macedonia, J.
Quatern. Sci., 25, 320–338, 2010b.
Vogel, H., Wessels, M., Albrecht, C., Stich, H. B., and Wagner, B.: Spatial variability of
recent sedimentation in Lake Ohrid (Albania/Macedonia), Biogeosciences, 7, 3333-3342,
2010c.
Wagner, B., Reicherter, K., Daut, G., Wessels, M., Matzinger, A., Schwalb, A., Spirkovski,
Z., and Sanxhaku, M.: The potential of Lake Ohrid for long-term palaeoenvironmental
reconstructions, Palaeogeogr. Palaeoclimatol. Palaeoecol., 259, 341-356, 2008a.
Wagner, B., Sulpizio, R., Zanchetta, G., Wulf, S., Wessels, M., Daut, G., and Nowaczyk, N.:
The last 40 ka tephrostratigraphic record of Lake Ohrid, Albania and Macedonia: a very distal



archive for ash dispersal from Italian volcanoes, J. Volcanol. Geotherm. Res., 177, 71-80,
2008b.
Wagner, B., Lotter, A. F., Nowaczyk, N., Reed, J. M., Schwalb, A., Sulpizio, R., Valsecchi,
V., Wessels, M., and Zanchetta, G.: A 40,000-year record of environmental change from
ancient Lake Ohrid (Albania and Macedonia), J. Paleolimnol., 41, 407-430, 2009.
Wagner, B., Vogel, H., Zanchetta, G., and Sulpizio, R.: Environmental changes within the
Balkan region during the past ca. 50 ka recorded in sediments form lakes Prespa and Ohrid,
Biogeosciences, 7,  2010.
Wagner, B., Francke, A., Sulpizio, R., Zanchetta, G., Lindhorst, K., Krastel, S., Vogel, H.,
Rethemeyer, J., Daut, G., Grazhdani, A., Lushaj, B., and Trajanovski, S.: Possible earthquake
trigger for 6[th] century mass wasting deposit at Lake Ohrid (Macedonia/Albania), Clim. Past,

15   8, 2069-2078, 2012.

Wagner, B., Wilke, T., Krastel, S., Zanchetta, G., Sulpizio, R., Reicherter, K., Leng, M. J.,
Grazhdani, A., Trajanovski, T., Francke, A., Lindhorst, K., Levkov, Z., Cvetkoska, A., Reed,
J., Zhang, X., Lacey, J., Wonik, T., Baumgarten, H., and Vogel, H.: The SCOPSCO drilling
project recovers more than 1.2 million history from Lake Ohrid, Sci. Drill., 17, 19-29, 2014.
Wijmstra, T. A.: Palynology of the first 30m of a 120m deep section in northern Greece, Acta
Bot. Neerl., 18, 511–527, 1969.
Wijmstra, T. A. and Smit, A.: Palynology of the middle part (30- 78 m) of a 120m deep
section in northern Greece (Macedonia), Acta Bot. Neerl., 25, 297–312, 1976.





Wilke, T., Albrecht, C., Anistratenko, V. V., Sahin, S. K., and Yildirim, Z.: Testing
biogeographical hypotheses in space and time: faunal relationships of the putative ancient
Lake Egirdir in Asia Minor, J. Biogeogr., 34, 1807–1821, 2007.
Wilke, T., Wagner, B., Albrecht, C., Ariztegui, D., Van Bocxlaer, B., Delicado, D., Francke,
A., Harzhauser, M., Hauffe, T., Holtvoeth, J., Just, J., Leng, M. J., Levkov, Z., Penkman, K.,
Sadori, L., Skinner, A., Stelbrink, B., Vogel, H., Wesselingh, F., and Wonik, T.: Scientific
drilling projects in ancient lakes: Integrating geological and biological histories, Glob. Planet.
Change, 143, 118-151, 2016.
Wolff, E. W., Chappellaz, J., Blunier, T., Rasmussen, S. O., and Svensson, A.: Millennial-
scale variability during the last glacial: The ice core record, Quaternary Sci. Rev., 29, 2828-

13    2838, 2010.

Wysocka, A., Grabowski, M., Sworobowicz, L., Mamos, T., Burzyński, A., and Sell, J.:
Origin of the Lake Ohrid gammarid species flock: ancient local phylogenetic lineage
diversification, J. Biogeogr., 41, 2014.
Zanchetta, G., Regattieri, E., Giaccio, B., Wagner, B., Sulpizio, R., Francke, A., Vogel, H.,
Sadori, L., Masi, A., Sinopoli, G., Lacey, J. H., Leng, M. L., Leicher, N.: Aligning and
synchronization of MIS5 proxy records from Lake Ohrid (FYROM) with independently dated
Mediterranean archives: implications for DEEP core chronology, Biogeosciences, 13, 2757-

23    2768, 2016.

Zhang, X. S., Reed, J. M., Lacey, J. H., Francke, A., Leng, M. J., Levkov, Z., Wagner, B.:
Complexity of diatom response to Lateglacial and Holocene climate and environmental
change in ancient, deep, and oligotrophic Lake Ohrid (Macedonia/Albania), Biogeosciences,

28    13, 1351-1365, 2016.



**Figure captions**
Figure 1: **(a)** Location of Lake Ohrid (black rectangle) on the Balkan Peninsula at the border
of the Former Yugoslav Republic of Macedonia (FYROM) and the Republic of Albania.
Other records mentioned in the text are indicated by red dots (core U1313 in the North
Atlantic, Sulmona basin in Italy, Tenaghi Philippon (TP) in Greece). **(b)** Map of the area of
lakes Ohrid and Prespa and bathymetric map of Lake Ohrid (from Lindhorst et al., 2015).
Coring locations of piston core Co1202 (red; Vogel et al., 2010) and ICDP sites (white) are
shown, with DEEP and Lini sites mentioned in the text. Secondary ICDP sites P (Pestani), G
(Gradiste), and C (Cerava) are not mentioned in the text. **(c)** Geological map of the Lake
Ohrid catchment (modified from Lindhorst et al., 2015).
Figure 2: Selected seismic profiles and calculated water depths at different times (see text for
details). The arrow of the reflector at 140 m blf (MIS 8/9) indicates the existence of a
secondary basin in the northwestern part of the lake. Please note that the lake was probably
larger for most periods but individual reflectors cannot be traced to the shallower water depth
due to faults. This also explains, why the estimated water depth is not zero at the edges of the
shown lake coverage.
Figure 3: Lithostratigraphy of the upper 247.8 mcd and tephra and crypto-tephra horizons in
the DEEP sediment sequence. For nomenclature and details see Leicher et al. (2016). Tephra
in bold was used as tie points for the age-depth model for the upper 247.8 mcd spanning the
last 637 kyr (Francke et al., 2016; Leicher et al., 2016). Tephrostratigraphic work on tephra
from below 247.8 mcd is ongoing.
Figure 4: Lake-level reconstructions (modified from Lindhorst et al., 2010, for details see
chapter 4.3.2, and this study), pollen (Sadori et al., 2016), sedimentological, and geochemical
data over the last 637 kyr (Francke et al., 2016; Just et al., 2016; Lacey et al., 2015) indicate a
long-term shift from cooler and wetter to drier and warmer glacial and interglacial periods
around 300 ka. Pollen curves have been corrected with respect to those reported in Sadori et
al. (2016). MIS boundaries are according to Lisiecki and Raymo (2005).





Figure 5: Geochemical data from the DEEP site sequence with sub-orbital changes during
MIS 12 in comparison with other records from a similar latitude (for location of North
Atlantic core U1313, the pollen record from Tenaghi Philippon and the isotope record from
Sulmona basin see Fig. 1). Arboreal pollen (AP) records are excluded of *Pinus*, *Juniperus*,
and *Betula* (Sadori et al., 2016); the record from Tenaghi Philippon is based on pollen data
from Wijmstra (1969) and Wijmstra and Smit (1976) and the age model from Tzedakis et al.
(2006); see also Sadori et al. (2016). Red bars and black dots at the bottom age axis indicate
tephrochronological tie points and tuning points used for the age model of the DEEP site
sequence (Francke et al., 2016).





Fig. 1





Fig. 2




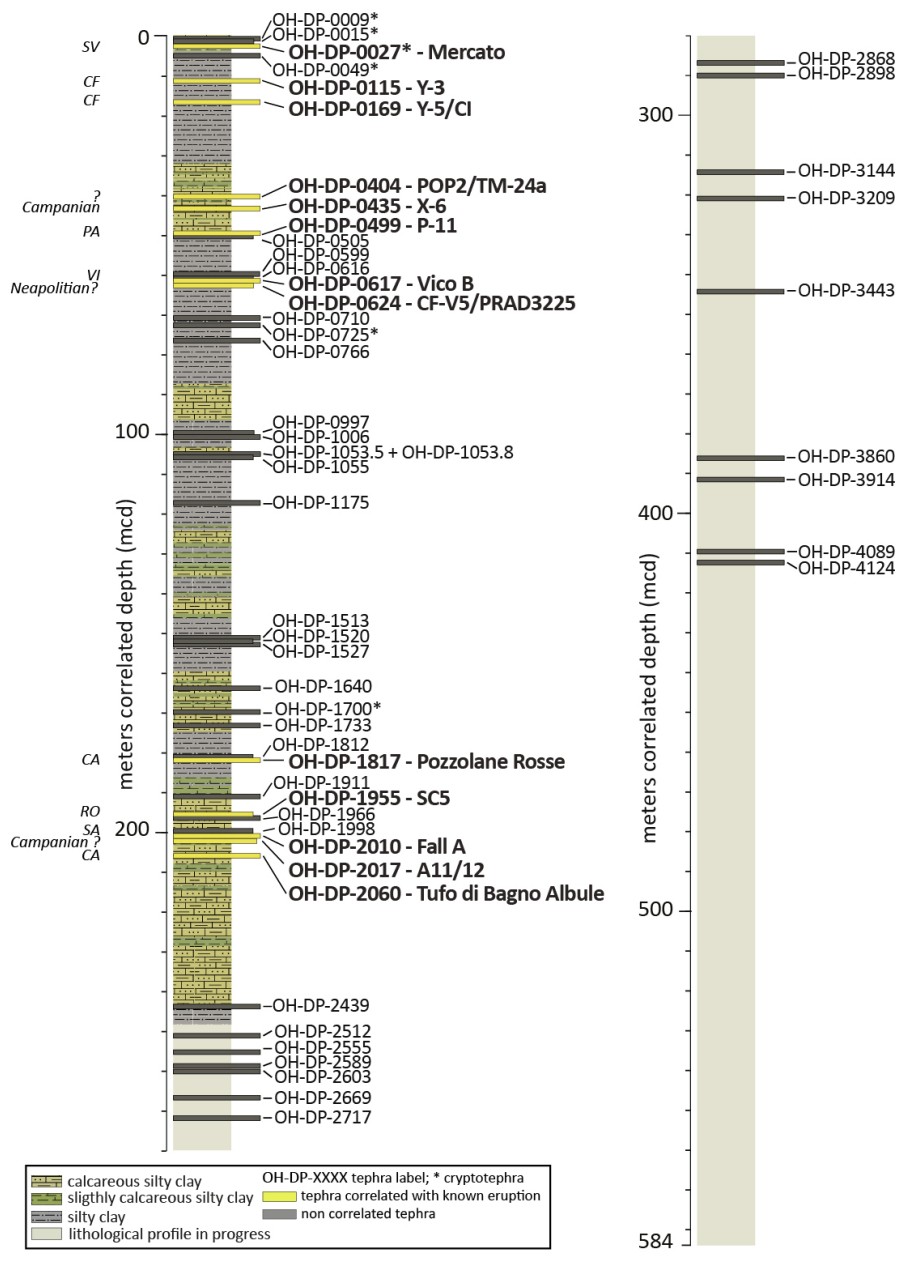

1      Fig. 3





Fig. 4



1    Fig. 5