# Peer review of "The environmental and evolutionary history of Lake Ohrid (FYROM/Albania): Interim results from the SCOPSCO deep drilling project"

_Biogeosciences, 2016_

## Referee Comment (RC1) · P.C. Tzedakis (Referee) · 16 Jan 2017

The MS presents a synthesis of initial results of the SCOPSCO deep drilling of Lake Ohrid project, previously published in a series of papers in Biogeosciences. It brings together information from the four main aims of the project (age and origin of lake; seis-motectonic history; volcanic activity and climate change; biodiversity and endemism) and compares results from different types evidence and approaches. As such, the whole is greater than the sum of its parts and the study is of great value to the scientific community. The text is well-written and organized and the figures of excellent quality. I

have one substantive comment and one minor quibble.

1. A potentially important conclusion emerging from several strands of evidence is a long-term trend from cooler and wetter to drier and warmer glacials and interglacials, starting at ∼300 ka. However, closer examination reveals that the trends between different types of evidence are not always congruous.

Water depths estimated from seismic data suggest a decrease in lake levels from 300 ka, but the trend is reversed from MIS 4 to today, with water depths increasing. The authors suggest (p. 17, l. 22) that this is in broad agreement with regional vegetation trends inferred from pollen analysis, but this not entirely accurate: the pollen data show that the two driest periods were the penultimate (MIS 6) and last glacial (MIS 4-2). This is mainly based on the large expansions of Artemisia during these intervals, vis-à-vis very low values in earlier glacials (incidentally, a feature that has not been observed in other long pollen sequences).

The claim that pollen data and inferred water depths show parallel trends is repeated on p. 20, but, again, if water depths increase from MIS 4, then there is divergence between the two over this interval. The pollen data suggest that in addition to glacials, a drying trend is also observed in interglacials. This is mainly based on the reduction of montane tree values in MIS 5 and MIS 1 (especially the almost complete disappearance of Picea; though Fagus increases somewhat from MIS 5c onwards) (Sadori et al., 2016). On the other hand, Mediterranean taxa percentages don't show any trend apart from a brief maximum at the MIS 4/3 boundary (which is unexpected), so it might be more useful to show the montane taxa in Fig. 4.

The drying trend theme is picked up again on p. 21, with the oxygen isotopic evidence. More specifically, a trend towards higher interglacial d18O in endogenic calcite after 300 ka is invoked, but close inspection shows that it is only MIS 5 that shows that; Holocene values are not that different from earlier interglacials. Interestingly, the d18O record from siderite shows lowest values in the penultimate and last glacials. This is

interpreted (p. 21, 30-31) as evidence for lower evaporation during glacials, which is reasonable. However, it is also interpreted as a "higher influence of winter precipitation (increased seasonality), which supports the interpretation of the palynological record". This, in fact, apears at odds with the high Artemisia expansions.

In conclusion, while the inference of a drying trend is potentially a very interesting and exciting observation, I would suggest that a more nuanced interpretation is needed, as close inspection reveals a more complicated picture amongst the different lines of evidence.

2. Referencing appears somewhat idiosyncratic at times, with an overall tendency to cite recent works. Thus, on p. 23 the attribution for the work on D/O and Heinrich events of the last glacial should include the original papers by Bond et al. (1992, 1993, Nature) and Dansgaard et al. (1993 Nature), while for older glacials McManus et al. (1999, Science), Raymo et al. (1998, Nature) and Barker et al. (2011, Science), probably deserve a mention. On the same page, (l. 12-13), important papers on the impact of HE and D/O events include Shackleton et al. (2000 Paleoceanography), Roucoux et al. (2001, QR), Margari et al. (2010 Nature Geoscience).

While the need to limit the overall number of references in a work of this wide scope is understandable, the paucity of references on the body of work on the environmental impacts of North Atlantic millennial-scale variability in the Balkans seems to be an oversight (e.g. Tzedakis et al., 2002, Science, 2004, Geology; Margari et al., 2009 QSR; Müller et al., 2011 QSR; Roucoux et al., 2011 JQS; Fletcher et al., 2013 QSR).

Finally, on the fascinating topic of the reservoir vs cradle function of Lake Ohrid, it might be worth recalling that that local buffering from extreme environmental effects in refugial areas may have not only led to reduced extinction rates, but also allowed lineage divergence to proceed, and thus refugia may have acted both as 'museums' for the conservation of diversity and as 'cradles' for the production of new diversity (Tzedakis et al., 2002 Science; Tzedakis, 2011 J. of Biogeogr.).

p. 14, l. 5 References for Lake Ioannina?

MPT: At several places mention is made of the 'end of the MPT'. Could this be more specific?

In sum, this is an extremely useful work and I am happy to recommend publication, subject to minor revision, which is needed to address the issues raised above.

---

## Referee Comment (RC2) · Anonymous Referee #1 · 8 Feb 2017

This paper present a significant body of new information on Lake Ohrid and its environmental history. I strongly support publication of this work. I look forward to a more careful discussion of iron in these sediments (Fe, Siderite variability) as part of the paleomagnetic chronostratigraphy and potential for early Fe sediment diagenesis.

---

## Author Comment (AC1) · 18 Feb 2017

Referee #1

*This paper presents a significant body of new information on Lake Ohrid and its environmental history. I strongly support publication of this work. I look forward to a more careful discussion of iron in these sediments (Fe, Siderite variability) as part of the paleomagnetic chronostratigraphy and potential for early Fe sediment diagenesis.*

We would like to thank referee #1 for the very positive review and we much appreciate his interest in the occurrence of diagenetic iron minerals. In order to meet the comment of referee #1, we will include some sentences with respect to the presence of Fe-sulfides, which is discussed in more detail in the recent paper of Just et al. (2016).

From the FTIRS and magnetic data it is evident that a shift of (early) diagenetic Fe-sulfides to siderite occurs around 320 ka. Notably these minerals predominantly occur in glacial sediments. Just et al. (2016) proposed that a change in sulfide availability, either by higher sulfate concentration in lake water or by upward migrating fluids, changed the geochemical regime in Lake Ohrid. Moreover, the occurrence of the siderites and Fe-sulfides in glacial sediments is likely related to higher Fe concentrations.

The, yet unpublished, data from the lower part of the core, imply that the glacial-interglacial variability of Fe-sulfides persists down to the base of the core. In consequence, polarity transitions are clear when they are located in interglacials, i.e., the Bruhnes/Matuyama and the base of the Jaramillo are very sharp. In contrast, the top of the Jaramillo is uncertain, and possibly duplicated by a later growth of greigite below the sediment surface. Analytical work is still ongoing and the paleomagnetic chronostratigraphy will be used along with tephrostratigraphic information and orbital tuning to establish an age model for the lower part of the sediment record, i.e. below 247.8 m composite depth or beyond ~640 ka. Moreover, we are currently performing sulfur isotope analyses on Lake Ohrid sediments, which will help to understand the change and source of sulfide concentrations. The age model for the entire DEEP site record and the results of the sulfur isotope analyses will be published in the near future in a separate paper.

Referee #2

*The MS presents a synthesis of initial results of the SCOPSCO deep drilling of Lake Ohrid project, previously published in a series of papers in Biogeosciences. It brings together information from the four main aims of the project (age and origin of lake; seismotectonic history; volcanic activity and climate change; biodiversity and endemism) and compares results from different types evidence and approaches. As such, the whole is greater than the sum of its parts and the study is of great value to the scientific community. The text is well-written and organized and the figures of excellent quality. I have one substantive comment and one minor quibble.*

We also would like to thank referee #2 for the very positive review and valuable suggestions.

*1. A potentially important conclusion emerging from several strands of evidence is a long-term trend from cooler and wetter to drier and warmer glacials and interglacials, starting at ~300 ka. However, closer examination reveals that the trends between different types of evidence are not always congruous.*

*Water depths estimated from seismic data suggest a decrease in lake levels from 300 ka, but the trend is reversed from MIS 4 to today, with water depths increasing. The authors suggest (p. 17, l. 22) that this is in broad agreement with regional vegetation trends inferred from pollen analysis, but this not entirely accurate: the pollen data show that the two driest periods were the penultimate (MIS 6) and last glacial (MIS 4-2). This is mainly based on the large expansions of Artemisia during these*

*intervals, vis-à-vis very low values in earlier glacials (incidentally, a feature that has not been observed in other long pollen sequences).*

*The claim that pollen data and inferred water depths show parallel trends is repeated on p. 20, but, again, if water depths increase from MIS 4, then there is divergence between the two over this interval. The pollen data suggest that in addition to glacials, a drying trend is also observed in interglacials. This is mainly based on the reduction of montane tree values in MIS 5 and MIS 1 (especially the almost complete disappearance of Picea; though Fagus increases somewhat from MIS 5c onwards) (Sadori et al., 2016). On the other hand, Mediterranean taxa percentages don't show any trend apart from a brief maximum at the MIS 4/3 boundary (which is unexpected), so it might be more useful to show the montane taxa in Fig. 4.*

*The drying trend theme is picked up again on p. 21, with the oxygen isotopic evidence. More specifically, a trend towards higher interglacial d18O in endogenic calcite after 300 ka is invoked, but close inspection shows that it is only MIS 5 that shows that; Holocene values are not that different from earlier interglacials. Interestingly, the d18O record from siderite shows lowest values in the penultimate and last glacials. This is interpreted (p. 21, 30-31) as evidence for lower evaporation during glacials, which is reasonable. However, it is also interpreted as a "higher influence of winter precipitation (increased seasonality), which supports the interpretation of the palynological record". This, in fact, appears at odds with the high Artemisia expansions.*

*In conclusion, while the inference of a drying trend is potentially a very interesting and exciting observation, I would suggest that a more nuanced interpretation is needed, as close inspection reveals a more complicated picture amongst the different lines of evidence.*

We completely agree with the referee that the inference of a drying trend with cooler and wetter to drier and warmer glacials and interglacials starting at ~300 ka is a very general statement and a more nuanced discussion with respect to the individual lines of evidence will help provide the nuance. A very detailed look at each proxy including comparisons is available in the individual papers, particularly in Lacey et al. (2016) and Sadori et al. (2016). As this new paper here is designed as an overview paper, we will try to keep the discussion more general, but will add relevant information to make differences more clearly.

- We stated already in the text that the hydro-acoustic data do not allow us to infer detailed and timely well-constrained lake level or climate changes due to tectonic activity and chronological uncertainties. This is evident as the new reconstruction supposes lowest lake level during MIS 5, whereas hydro-acoustic and sediment core data in Lindhorst et al. (2010) infer lowest lake level during early MIS 6. We will re-check the text, if this needs to be pointed out more clearly.

- The pollen data show an unexpected behavior of *Artemisia*. High amounts of other herbs (grasses, chenopods, Cichorioideae and Cyperaceae) are found already in MIS 10/12 glacials and indicate the presence of open formations, even if less dry than those characterized by *Artemisia*. The remarkably high amount of *Pinus* pollen grains in specific intervals (i.e. MIS 9 to 12) of the skeleton DEEP Lake Ohrid pollen diagram was already identified as an issue in Sadori et al. (2016), which led to the decision to exclude pines from the pollen sum. We are confident that the basic signal produced is reliable, but we will carefully cross-check sample processing. The low *Fagus* percentages are described in more detail in Sadori et al. (2016) and we agree with referee #2 that there is a slight increase in *Fagus*, when spruce decreases, which is probably the result of a rearrangement of vegetation in altitudinal belts.
We also figured out that there was an error in the drawing of the Mediterranean taxa and the pioneers curve. We will submit a corrigendum to the Sadori et al. (2016) paper. As the short peak of Mediterranean taxa at the MIS 4/3 boundary resulted from the error in the drawing

and this curve is not so significant for a site such Ohrid, we will remove this curve from Fig. 4 and use the corrected drawing (see below).

[Figure]

Figure 4 of the BGD paper, modified according to the discussion in the text.

- We do not agree with the comment that only MIS 5 shows a drying trend. Lacey et al. (2016) show a comparison figure of average d18O values, which clearly shows MIS 5 and 1 have the highest average baseline of the last 640 ka. Also, when calcite data or "warm periods" (rather than inter glacials senso stricto) are considered, there is a clear drying trend

through from 300 ka after the transition to lower d18O in MIS 9. The isotope data thus support the very general lake level reconstruction based on the hydro-acoustic data with lowest lake levels during MIS 6 or 5. We agree with referee #2, however, that the sentence "higher influence of winter precipitation (increased seasonality), which supports the interpretation of the palynological record" needs to be removed, as the two proxies do not show an unequivocal pattern of seasonality, thus confirming the need for a more nuanced interpretation and discussion.

*2. Referencing appears somewhat idiosyncratic at times, with an overall tendency to cite recent works. Thus, on p. 23 the attribution for the work on D/O and Heinrich events of the last glacial should include the original papers by Bond et al. (1992, 1993, Nature) and Dansgaard et al. (1993 Nature), while for older glacials McManus et al. (1999, Science), Raymo et al. (1998, Nature) and Barker et al. (2011, Science), probably deserve a mention. On the same page, (l. 12-13), important papers on the impact of HE and D/O events include Shackleton et al. (2000 Paleoceanography), Roucoux et al. (2001, QR), Margari et al. (2010 Nature Geoscience).*

*While the need to limit the overall number of references in a work of this wide scope is understandable, the paucity of references on the body of work on the environmental impacts of North Atlantic millennial-scale variability in the Balkans seems to be an oversight (e.g. Tzedakis et al., 2002, Science, 2004, Geology; Margari et al., 2009 QSR; Müller et al., 2011 QSR; Roucoux et al., 2011 JQS; Fletcher et al., 2013 QSR). Finally, on the fascinating topic of the reservoir vs cradle function of Lake Ohrid, it might be worth recalling that that local buffering from extreme environmental effects in refugial areas may have not only led to reduced extinction rates, but also allowed lineage divergence to proceed, and thus refugia may have acted both as 'museums' for the conservation of diversity and as 'cradles' for the production of new diversity (Tzedakis et al., 2002 Science; Tzedakis, 2011 J. of Biogeogr.).*

We tried to keep the number of references reasonable and therefore did not include important papers, which also deserve to be cited in the text. Following the suggestion of referee #2, we will add relevant papers and information.

*p. 14, l. 5 References for Lake Ioannina?*

Lindhorst et al. (2015) refer to Tzedakis (1994). We will add this reference in the text.

*MPT: At several places mention is made of the 'end of the MPT'. Could this be more specific?*

For the MPT, we will refer in the text to the period between 1250 and 700 ka according to Clark et al. (2006).

*In sum, this is an extremely useful work and I am happy to recommend publication, subject to minor revision, which is needed to address the issues raised above.*

Thank you !

References

Clark, P. U., Archer, D., Pollard, D., Blum, J. D., Rial, J. A., Brovkin, V., Mix, A. C., Pisias, N. G., and Roy, M.: The middle Pleistocene transition: characteristics, mechanisms, and implications for long-term changes in atmospheric pCO2, Quaternary Sci. Rev., 25, 3150–3184, 2006.

Lacey, J. H,, Leng, M. J., Francke, A., Sloane, H. J., Milodowski, A., Vogel, H., Baumgarten, H., Zanchetta, G., and Wagner, B.: Northern Mediterranean climate since the Middle Pleistocene: a 637 ka stable isotope record from Lake Ohrid (Albania/Macedonia), Biogeosciences, 13, 1801-1820, 2016.

Lindhorst, K., Vogel, H., Krastel, S., Wagner, B., Hilgers, A., Zander, A., Schwenk, T., Wessels, M., and Daut, G.: Stratigraphic analysis of lake level fluctuations in Lake Ohrid: An integration of high resolution hydro-acoustic data and sediment cores, Biogeosciences, 7, 3531-3548, 2010.

Lindhorst, K., Krastel, S., Reicherter, K., Stipp, M., Wagner, B., and Schwenk, T.: Sedimentary and tectonic evolution of Lake Ohrid (Macedonia/Albania), Basin Research, 27, 84-101, 2015.

Sadori, L., Koutsodendris, A., Panagiotopoulos, K., Masi, A., Bertini, A., Combourieu-Nebout, N., Francke, A., Kouli, K., Joannin, S., Mercuri, A. M., Peyron, O., Torri, P., Wagner, B., Zanchetta, G., Sinopoli, G., Donders, T. H.: Pollen-based paleoenvironmental and paleoclimatic change at Lake Ohrid (SE Europe) during the past 500 ka, Biogeosciences,13, 1423-1437, 2016.

Tzedakis, P.: Vegetation change through glacial-interglacial cycles: a long pollen sequence perspective, Phil. Trans. R. Soc. B, 345, 403–432, 1994.

---

## Author Response (AR1)

Referee #1

*This paper presents a significant body of new information on Lake Ohrid and its environmental history. I strongly support publication of this work. I look forward to a more careful discussion of iron in these sediments (Fe, Siderite variability) as part of the paleomagnetic chronostratigraphy and potential for early Fe sediment diagenesis.*

In order to meet the comment of referee #1, we included in chapter 4.3.2 "Environmental history - Long-term changes" some sentences with respect to the presence of Fe-sulfides, which is discussed in more detail in the recent paper of Just et al. (2016).

".... Conversely, drier and warmer conditions after ca. 320 ka likely reduced mixing of the water column during the interglacials, which would lead to anoxic bottom waters and a better preservation of organic matter. Just et al. (2016) proposed a decrease in sulfide availability, either by lower sulfate concentration in lake water or ceased upward migrating fluids, changing the geochemical regime in Lake Ohrid. Such conditions are indicated by a shift from predominant glacial formation of Fe-sulfides to siderite around 320 ka, when higher Fe concentrations and limited sulphur content of sediments may have prevented the formation of greigite (Fig. 4; Just et al., 2016)."

Referee #2

*The MS presents a synthesis of initial results of the SCOPSCO deep drilling of Lake Ohrid project, previously published in a series of papers in Biogeosciences. It brings together information from the four main aims of the project (age and origin of lake; seismotectonic history; volcanic activity and climate change; biodiversity and endemism) and compares results from different types evidence and approaches. As such, the whole is greater than the sum of its parts and the study is of great value to the scientific community. The text is well-written and organized and the figures of excellent quality. I have one substantive comment and one minor quibble.*

*1. A potentially important conclusion emerging from several strands of evidence is a long-term trend from cooler and wetter to drier and warmer glacials and interglacials, starting at ~300 ka. However, closer examination reveals that the trends between different types of evidence are not always congruous.*

*Water depths estimated from seismic data suggest a decrease in lake levels from 300 ka, but the trend is reversed from MIS 4 to today, with water depths increasing. The authors suggest (p. 17, l. 22) that this is in broad agreement with regional vegetation trends inferred from pollen analysis, but this not entirely accurate: the pollen data show that the two driest periods were the penultimate (MIS 6) and last glacial (MIS 4-2). This is mainly based on the large expansions of Artemisia during these intervals, vis-à-vis very low values in earlier glacials (incidentally, a feature that has not been observed in other long pollen sequences).*

*The claim that pollen data and inferred water depths show parallel trends is repeated on p. 20, but, again, if water depths increase from MIS 4, then there is divergence between the two over this interval. The pollen data suggest that in addition to glacials, a drying trend is also observed in interglacials. This is mainly based on the reduction of montane tree values in MIS 5 and MIS 1 (especially the almost complete disappearance of Picea; though Fagus increases somewhat from MIS 5c onwards) (Sadori et al., 2016). On the other hand, Mediterranean taxa percentages don't show any trend apart from a brief maximum at the MIS 4/3 boundary (which is unexpected), so it might be more useful to show the montane taxa in Fig. 4.*

*The drying trend theme is picked up again on p. 21, with the oxygen isotopic evidence. More specifically, a trend towards higher interglacial d18O in endogenic calcite after 300 ka is invoked, but close inspection shows that it is only MIS 5 that shows that; Holocene values are not that different from earlier interglacials. Interestingly, the d18O record from siderite shows lowest values in the penultimate and last glacials. This is interpreted (p. 21, 30-31) as evidence for lower evaporation during glacials, which is reasonable. However, it is also interpreted as a "higher influence of winter precipitation (increased seasonality), which supports the interpretation of the palynological record". This, in fact, appears at odds with the high Artemisia expansions.*

*In conclusion, while the inference of a drying trend is potentially a very interesting and exciting observation, I would suggest that a more nuanced interpretation is needed, as close inspection reveals a more complicated picture amongst the different lines of evidence.*

We completely agree with the referee that the inference of a drying trend with cooler and wetter to drier and warmer glacials and interglacials starting at ~300 ka is a very general statement and a more nuanced discussion with respect to the individual lines of evidence will help provide the nuance. A very detailed look at each proxy including comparisons is available in the individual papers, particularly in Lacey et al. (2016) and Sadori et al. (2016). As this new paper here is designed as an overview paper, we kept the discussion more general, but added relevant information to emphasize differences more clearly. In order to meet the comment we modified also the respective sentence in the abstract.
" The multi-proxy dataset covering these 637 kyr indicates long-term variability. Some of the proxies show a general trend from cooler and wetter to drier and warmer glacial and interglacial periods around 300 ka."

- We stated already in the text that the hydro-acoustic data do not allow us to infer detailed and timely well-constrained lake level or climate changes due to tectonic activity and chronological uncertainties. This is evident as the new reconstruction supposes lowest lake level during MIS 5, whereas hydro-acoustic and sediment core data in Lindhorst et al. (2010) infer lowest lake level during early MIS 6. We added the following sentences (chapter 4.3.2 "Environmental history - Long-term changes") to point out the differences between the two reconstructions:
" The lake level curve from north-eastern Lake Ohrid is only partly in phase with the minimum lake level curve based on the new hydro-acoustic reconstruction (Figs 2 and 4). Whereas the terraces in the northeastern basin provide relatively precise water depths, the reconstruction based on hydro-acoustic information (Fig. 2) can provide only minimum water depths and is certainly biased by subsidence."

- We modified the respective sentences to changes in pollen community in chapter 4.3.2 "Environmental history - Long-term changes"). Following the suggestion of the reviewer, we replaced Mediterranean taxa in Fig. 4 montane taxa and added relevant information in this chapter.
We also figured out that there was an error in the drawing of the Mediterranean taxa and the pioneers curve. We will submit a corrigendum to the Sadori et al. (2016) paper. As the short peak of Mediterranean taxa at the MIS 4/3 boundary resulted from the error in the drawing and this curve is not so significant for a site such Ohrid, we removed this curve from Fig. 4 and used the corrected drawing (new Fig. 4).

- We do not agree with the comment that only MIS 5 shows a drying trend. Lacey et al. (2016) show a comparison figure of average $\partial^{18}O$ values, which clearly shows MIS 5 and 1 have the highest average baseline of the last 640 ka. Also, when calcite data or "warm periods" (rather than inter glacials senso stricto) are considered, there is a clear drying trend through from 300 ka after the transition to lower $\partial^{18}O$ in MIS 9. The isotope data thus support the very general lake level reconstruction based on the hydro-acoustic data with lowest lake levels during MIS 6 or 5. We agree with referee #2, however, that the sentence "higher influence of winter precipitation (increased seasonality), which supports the interpretation of the palynological record" needs to be removed, as the two proxies do not show an unequivocal pattern of seasonality, thus confirming the need for a more nuanced interpretation and discussion.

- We added some sentences to the higher late MIS 6 climate variability, which matches well with the record from Lake Ioannina.
"... It thus seems that the outlet was active most of the time and climate driven lake-level change may have existed only for relatively short periods or has been compensated at least partly by tectonic activity. Significant variations in TOC and isotope data during early MIS 6 imply a higher variability of the climate compared to the latter period of MIS 6 (Fig. 4). These observations correspond well with palynological studies from the Ioannina basin, where distinct vegetation changes between 185-155 ka indicate a high climate variability, whereas a greater abundance of steppe taxa and other herbaceous elements, combined with lower tree pollen percentages, during the latter MIS 6 after 155 ka indicate that the landscape was predominantly open in character and more stable (Roucoux et al., 2011)."

*2. Referencing appears somewhat idiosyncratic at times, with an overall tendency to cite recent works. Thus, on p. 23 the attribution for the work on D/O and Heinrich events of the last glacial should include the original papers by Bond et al. (1992, 1993, Nature) and Dansgaard et al. (1993 Nature), while for older glacials McManus et al. (1999, Science), Raymo et al. (1998, Nature) and Barker et al. (2011, Science), probably deserve a mention. On the same page, (l. 12-13), important papers on the impact of HE and D/O events include Shackleton et al. (2000 Paleoceanography), Roucoux et al. (2001, QR), Margari et al. (2010 Nature Geoscience).*

*While the need to limit the overall number of references in a work of this wide scope is understandable, the paucity of references on the body of work on the environmental impacts of North Atlantic millennial-scale variability in the Balkans seems to be an oversight (e.g. Tzedakis et al., 2002, Science, 2004, Geology; Margari et al., 2009 QSR; Müller et al., 2011 QSR; Roucoux et al., 2011 JQS; Fletcher et al., 2013 QSR). Finally, on the fascinating topic of the reservoir vs cradle function of Lake Ohrid, it might be worth recalling that that local buffering from extreme environmental effects in refugial areas may have not only led to reduced extinction rates, but also allowed lineage divergence to proceed, and thus refugia may have acted both as 'museums' for the conservation of diversity and as 'cradles' for the production of new diversity (Tzedakis et al., 2002 Science; Tzedakis, 2011 J. of Biogeogr.).*

We tried to keep the number of references reasonable and therefore did not include important papers, which also deserve to be cited in the text. Following the suggestion of referee #2, we added all relevant papers and information.

*p. 14, l. 5 References for Lake Ioannina?*

Lindhorst et al. (2015) refer to Tzedakis (1994). We added this reference in the text.

*MPT: At several places mention is made of the 'end of the MPT'. Could this be more specific?*

For the MPT, we now refer in the text to the period between 1250 and 700 ka according to Clark et al. (2006).

*In sum, this is an extremely useful work and I am happy to recommend publication, subject to minor revision, which is needed to address the issues raised above.*

We added the acknowledgements and included our thanks to both reviewers.

[revised manuscript text omitted]